# INSTRUCTIONGPT-4: A 200-INSTRUCTION PARADIGM FOR FINE-TUNING MINIGPT-4

## ABSTRACT

Multimodal large language models are typically trained in two stages: first pre-training on image-text pairs, and then fine-tuning using supervised vision-language instruction data. Recent studies have shown that large language models can achieve satisfactory results even with a limited amount of high-quality instruction-following data. In this paper, we introduce InstructionGPT-4, which is fine-tuned on a small dataset comprising only 200 examples, amounting to approximately 6% of the instruction-following data used in the alignment dataset for MiniGPT-4 (Zhu et al., 2023). To achieve this, we first propose several metrics to access the quality of multimodal instruction data. Based on these metrics, we present an effective and trainable data selector to automatically identify and filter low-quality vision-language data. By employing this method, InstructionGPT-4 outperforms the original MiniGPT-4 on various evaluations. Overall, our findings demonstrate that less but high-quality instruction tuning data is efficient in enabling multimodal large language models to generate better output.

## 1 INTRODUCTION

GPT-4 (OpenAI, 2023) has showcased its powerful prowess in generating highly detailed and precise descriptions of images, signaling a new era of language and visual processing. Thus, GPT-4 like Multimodal Large Language Models (MLLMs) have recently emerged as a prominent research area, harnessing powerful Large Language Models (LLMs) as a cognitive framework for conducting multimodal tasks. The remarkable and unexpected capabilities exhibited by MLLMs surpass those of traditional methods, indicating a potential pathway towards artificial general intelligence. To achieve this, massive image-text pairs and vision-language instruction tuning data have been employed to train simple connectors (e.g., MiniGPT-4 (Zhu et al., 2023) and LLaVA (Liu et al., 2023b)) between frozen LLMs (e.g., LLaMA (Touvron et al., 2023a) and Vicuna (Chiang et al., 2023)) and visual representations (e.g., CLIP (Radford et al., 2021) and BLIP-2 (Li et al., 2023b)).

MLLMs are usually trained in two stages: pre-training and fine-tuning (Zhu et al., 2023; Liu et al., 2023b; Gao et al., 2023; Dai et al., 2023). Pre-training on image-text pairs helps MLLMs gain a large amount of knowledge while fine-tuning teaches models to better understand human intentions and generate accurate responses. Recently, instruction tuning on large-scale datasets has served as a powerful fine-tuning technique to empower MLLMs with enhanced vision-language understanding and instruction-following abilities (Zhao et al., 2023; Zhang et al., 2023b; Liu et al., 2023a). It facilitates the alignment of models with human preferences, enabling the generation of desired outputs in response to various instructions. Recent state-of-the-arts, including InstructBLIP (Dai et al., 2023) and Otter (Li et al., 2023a), have shown promising results by leveraging a collection of vision-language datasets for visual instruction tuning.

However, it has been observed that commonly used instruction-tuning datasets surprisingly contain numerous low-quality instances with incorrect or irrelevant responses (Zhou et al., 2023; Chen et al., 2023; Cao et al., 2023). Such data can mislead and negatively impact the performance of the model. This issue has prompted researchers to delve into the possibility of achieving robust performance using a small quantity of high-quality instruction-following data. Encouragingly, recent studies have substantiated the promising potential of this approach. Zhou et al. (2023) introduce LIMA, a language model fine-tuned with carefully curated high-quality data, selected by human experts.

This study has shown that LLMs can achieve satisfactory results even with a limited amount of high-quality instruction-following data. The proposed idea "Less is More" tells that data quality is more important than data quantity to improve model performance, which does not conflict with the Scaling Law (Kaplan et al., 2020). Building upon these foundations, our objective is to determine if using less instruction data can yield better alignment results in multimodal large language models. Nevertheless, there is a challenge that the process of collecting appropriate high-quality vision-language datasets for fine-tuning multimodal language models lacks clear guidelines.

Different from LIMA (Zhou et al., 2023) that requires manually constructed dataset, we aim to propose a robust and effective data selector that automatically identifies and filters low-quality vision-language data from existing datasets, ensuring that our model is trained on the most relevant and informative samples. The key focus of our study lies in exploring the efficacy of reduced but high-quality instruction-tuning data in fine-tuning MLLMs. Another challenge is the current lack of comprehensive methods for evaluating the quality of vision-language data. Therefore, we introduce several novel metrics tailored for assessing the quality of multimodal instruction data, including CLIP Score (Radford et al., 2021), GPT Score (Chen et al., 2023), Reward Score (OpenAssistant, 2023), Length Score and Multimodal Features of each vision-language data.

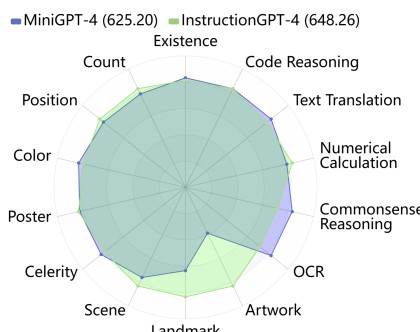

Figure 1: Comparison of MME evaluation (InstructionGPT-4 vs. MiniGPT-4).

To investigate the metrics' relationship with the real instruction data quality, we first split a range of distinct subsets from the original fine-tuning data. Subsequently, we record each fine-tuned model's performance on the validation set as the labels of data quality. We then compute the metrics and multimodal data features and combine them as an embedding across each subset. After that, we apply a self-attention network as the data selector to determine the relationship between the genuine quality labels and embeddings. We perform spectral clustering (Ng et al., 2001), which aims to ensure the diversity of data distribution, on the original 3.4K data used to fine-tune MiniGPT-4. Finally, we apply the data selector on each cluster to predict its quality label and sort. Through this series of procedures, InstructionGPT-4 is fine-tuned on a much smaller but carefully selected subset following the same training configuration of MiniGPT-4.

Our evaluations focus on a wide range of complex open-domain multimodal large language model benchmarks, including MME (Fu et al., 2023), MMBench (Liu et al., 2023c), VQA datasets from LVLM-eHub (Xu et al., 2023), etc. Through rigorous experimentation, we demonstrate that 200 pieces of data used for fine-tuning, which is 6% of the original scale, are enough to help InstructionGPT-4 achieve comprehensive superiority over MiniGPT-4 across these diverse multimodal tasks, with a +23 score enhancement on MME, a +1.55 score improvement on MMBench, and a +1.76% increase in performance on VQA datasets compared to MiniGPT-4. Specifically, InstructionGPT-4 outperforms MiniGPT-4 in 8 out of 14 tasks within MME, 13 out of 20 abilities in MMBench, and excels in all four VQA datasets included in LVLM-eHub. This discovery is inspiring, as it demonstrates that the data quality in vision-language instruction tuning can outweigh the quantity. In addition, this shift towards prioritizing data quality presents a new and more efficient paradigm that can generally improve the fine-tuning stage of MLLMs.

Our contributions are summarized as follows:

- We are the first to demonstrate that less instruction data for better alignment is also suitable for multimodal large language model, by showing that fine-tuning MiniGPT-4 with only 6% instruction-following data can achieve better performance.

- We introduce the concept of genuine quality labels along with a set of indicators for evaluating the quality of multimodal instruction-following data, and propose a learnable data selector to obtain high-quality vision-language data for fine-tuning.

- Our InstructionGPT-4 fine-tuned with 200 instructions consistently outperforms the original MiniGPT-4 in various popular benchmarks such as MME, MMBench and VQA datasets.

| Indicators | Explanation |
|---|---|
| CLIP Score | The cosine similarity between image embedding and response text embedding. The CLIP Score serves as a measure of the alignment between the provided image and its accompanying caption. This score quantifies how well the caption accurately describes the visual content, ensuring that the image and text are in concordance. |
| Length Score | The length of every answer in the multimodal dataset. The length metric gauges the extent of information encapsulated within the caption. A balanced and informative answer length is crucial to convey the desired instruction without being excessively verbose or overly concise. |
| Reward Score | Score from a reward model (OpenAssistant, 2023) that judges the human likeness to a response. The reward model is trained from human feedback to predict which generated answer is better judged by a human, given a question. |
| GPT Score | Score from GPT4 (OpenAI, 2023) to evaluate the quality of response. The GPT Score reflects the LLM's assessment of the caption's quality. This score is indicative of how effectively the generated caption adheres to the model's language proficiency, considering factors such as grammar, semantics, and fluency. |
| Multimodal Features | Vision-language features in low dimensional space obtained by encoding images with ViT from CLIP (Radford et al., 2021) and text with Llama2 (Touvron et al., 2023b), followed by conducting unsupervised dimensionality reduction. |

Table 1: Quantitative indicators and explanations for evaluating instruction-following data quality. CLIP Score measures the suitability between the image and caption. Length Score, Reward Score, and GPT Score measure the comprehensive quality of the caption. Multimodal Features represent the essential characteristics of vision-language data.

## 2 INSTRUCTIONGPT-4

In this paper, we present InstructionGPT-4, a multimodal large language model fine-tuned on a set of 200 high-quality instructions carefully chosen from the dataset utilized in the second-stage training (comprising 3.4K instructions) of MiniGPT-4 (Zhu et al., 2023). The core of InstructionGPT-4 is the selection of high-quality instructions. In particular, we want to find out if there exists a subset making InstructionGPT-4 achieve better performance. Thus, in this section, we begin by defining genuine quality labels and presenting indicators for assessing the quality of multimodal instruction-following data. Subsequently, we train a learnable data selector to align these indicators with the genuine labels. An overall procedure of the data selector is illustrated in Figure 2.

### 2.1 SELECTING PRINCIPLE

Selecting useful multimodal instruction data is crucial for effectively training MLLMs. Following LIMA (Zhou et al., 2023), we propose two key principles for selecting optimal instruction data: diversity and quality.

**Diversity.** As most of the knowledge is obtained during the pre-training stage for MLLMs, it is necessary to gain better alignment abilities by training on diverse vision-language instruction data. We adopt spectral clustering on the image embeddings encoded to divide the data into ten categories. Our ablation study is detailed in Section 4.4.

**Quality.** Vision-language instruction data teaches the multimodal model to follow a certain pattern when interacting with users. Hence, the quality of these instruction-following data could be viewed as its ability to efficiently steer multimodal language models in learning to generate responses in a particular manner closely related to the instruction data style. In Section 2.2, we present our multimodal instruction selection process. We introduce the concept of genuine quality labels along with several related indicators designed for a quantitative assessment of data quality. The specific indicators used for this quantitative evaluation of data quality are outlined in Table 1, while the genuine quality labels are presented in Table 7.

### 2.2 INDICATORS AND GENUINE QUALITY LABELS

Inspired by Instruction Mining (Cao et al., 2023) which estimates the data quality by the loss produced by the fine-tuned model, we propose that assessing the real quality of a set of data is contingent on its effectiveness in training a model, i.e., whether the model performs well when trained on this dataset. Therefore, we assert that the metrics (e.g., accuracy, F1 score (Sasaki et al., 2007)) obtained

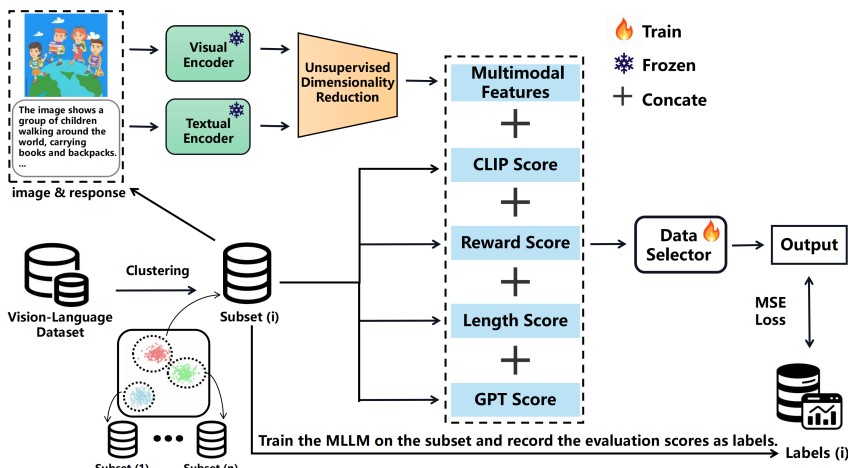

Figure 2: Overall procedures of the data selector. We first split the vision-language dataset into $n$ subsets. Subsequently, we train the MLLM on each subset and record the evaluation scores as genuine quality labels. Additionally, we concatenate various indicators generated from these subsets to form embeddings. These embeddings were then used to train the data selector, with the objective of aligning the embeddings with the quality labels.

when evaluating the model after training on this dataset can be considered genuine labels for evaluating the quality of this dataset. However, training an MLLM for evaluation in various datasets can be inefficient. To conveniently assure the quality of the selected multimodal instruction data, we formulate a set of indicators for assessment in Table 1 and train a neural network as a data selector to fit the indicators to the genuine quality labels. Thus, the data selector can be applied to other different multimodal datasets directly. Here we introduce how to obtain genuine quality labels and indicators for vision-language data.

Given a small set of vision-language instruction data $D$, it is used to fine-tune a pre-trained MLLM, and the fine-tuned model is subsequently evaluated on a series of validation datasets to obtain an average score, which serves as a genuine quality label $y_D$ in Appendix A.2 for the set.

For the triplets $x$ containing images, instructions, and responses in $D$, we employ CLIP Score (Radford et al., 2021) $C(x)$ to measure the matching degree between the image and the response. We also apply the length of responses $L(x)$, and take the Reward Score (OpenAssistant, 2023) into consideration, which is $R(x)$. We prompt GPT-4 (OpenAI, 2023) as an auto-grader rating each sample $x \in D$ with a GPT Score $G(x, p_G)$ wherein $p_G$ is the rating instruction that is designed based on the prompt from Alpagasus (Chen et al., 2023) shown in Appendix A.1. These four scores in Table 1 are intuitive and clear, and they can more completely cover the various aspects of multimodal data quality. Using a single score to filter data can be useful, but it may not provide a comprehensive measure of data quality. Therefore, it is necessary to combine multiple indicators as an embedding to assess data quality collectively. By concatenating the four scores with the image-text features, the created embeddings can more comprehensively represent the characteristics of multimodal data. The high dimensionality of image-text features, which come from a frozen visual encoder $f$ (e.g., ViT (Ilharco et al., 2021)) and a frozen textual encoder $g$ (e.g., Llama2 (Touvron et al., 2023b)), can indeed lead to a large number of parameters that need to be learned during the fitting process, making the task potentially prone to overfitting. Using an unsupervised dimensionality reduction method $P$ (e.g., Principal Component Analysis) separately to the image and text features that can preserve important information without training is a sensible approach to address this issue. Consequently, each piece of multimodal data can be assigned an embedding $e$ based on these indicators in Table 1, i.e.,

$$e(x) = \text{Concat}\left[C(x), L(x), R(x), G(x, p_G), P(f(x_{\text{image}}), g(x_{\text{response}}))\right]. \tag{1}$$

Our framework is general and not limited to these indicators. Other metrics to measure the quality of multimodal data can also be considered, such as perplexity and Error L2-Norm (Paul et al., 2021). We leave exploring possibly more effective and sophisticated architectural designs as future work.

## 2.3 DATA SELECTOR

**Training.** Given a vision-language instruction dataset, a reasonable and straightforward strategy to obtain the genuine quality labels is to divide the original multimodal dataset into $n$ subsets of equal size through clustering (e.g., $K$-means++). For each subset $i$, we now obtain the embedding of every triplet in subset $i$, along with the quality label $y_i$, as detailed in Section 2.2. We combine these embeddings into a single composite batch denoted as $e_i$, paired with its corresponding quality label $y_i$. Having gathered a collection of such pairs $(e_i, y_i)$ from all $n$ subsets, we can then proceed to learn a data selector $F$ to fit the batches $\{e_i\}_{i=1}^n$ to quality labels $\{y_i\}_{i=1}^n$. The data selector could take various forms, such as a linear layer, an MLP, or a self-attention.

**Testing.** Given a multimodal dataset $D$ of triplets $x$ = (image, instruction, answer) with $x \in D$ and a pre-trained MLLM (e.g., MiniGPT-4), our ultimate objective is to identify a high-quality subset $S \subset D$ that, when utilized for fine-tuning, leads to the improvement of the pre-trained MLLM.

In order to select $S$ from $D$ and ensure its diversity, we first use a clustering algorithm (e.g., spectral clustering) to separate the images in $D$ into $K$ groups. The clustering algorithm is supposed to be different from the previous one because each of the former clusters shares the same quality label. Suppose that the total amount of $D$ is $|D|$ and the $i$-th group's amount is $|D_i|$. We set $|S| = \alpha$ as the size of the target subset.

For each $x$ in $D$, we gain an embedding $e(x)$ in Equation equation 1. We sort $x$ according to the predicted label $F(e(x))$ and select $S_i$ from each group $D_i$. Each $S_i$ contains top $|S_i|$ triplets $x$ based on $F(e(x))$ from $D_i$, i.e.,

$$|S_i| = \frac{\alpha \cdot |D_i|}{|D|}, \quad S_i = \arg \max_{V \subset D_i, |V|=|S_i|} \sum_{x \in V} F(e(x)) \tag{2}$$

At last, we combine these $K$ subgroups:

$$S = S_1 \cup S_2 \cup \ldots \cup S_K, \tag{3}$$

where $S$ is the final high-quality dataset selected by the data selector. The whole and detailed selection procedure in the training testing stage is shown in Appendix A.3.

## 3 EXPERIMENTAL SETUP

### 3.1 IMPLEMENTATION DETAILS

Our data selector training and subsequent data selection are both on the cc_sbu_align dataset (Zhu et al., 2023), which is used for the second stage fine-tuning in MiniGPT-4 and contains 3439 triplets comprising instructions, images, and responses.

For the training process, we apply the $K$-means++ (Arthur & Vassilvitskii, 2007) to split the vision-language dataset into 30 subsets, each containing 114 data points, for acquiring genuine quality labels (detailed in Table 7). This subset count balances sufficient samples for data selector training and good alignment results for the multimodal model on each subset. Furthermore, $K$-means++ is employed to ensure that indicators within each subset are similar and differ between subsets. This division strategy guarantees label differentiation for each subset when adjusting the indicators, aiding in data quality assessment. To acquire genuine quality labels (which necessitate training a multimodal model on each cluster group for evaluation) without letting the quantity of data impact the capabilities of the trained multimodal models, we employ a simple post-processing technique in $K$-means++ clustering. This involves identifying clusters with either excessively high or low sample counts, and redistributing some samples from the larger clusters to the smaller ones based on their distance to different cluster centroids, thereby equalizing the number of samples in each cluster. The data selector is implemented using a self-attention architecture, comprising 2 layers with residual connections. The size of multimodal features concated in the embedding is set to 6. The size of the final subset $S$ selected by the data selector is set to $\alpha = 200$, which contains 6% of the original vision-language instruction data. Each fine-tuned model is evaluated on the evaluation dataset mentioned in Section 3.2. More experimental setting details can be found in Appendix A.4.

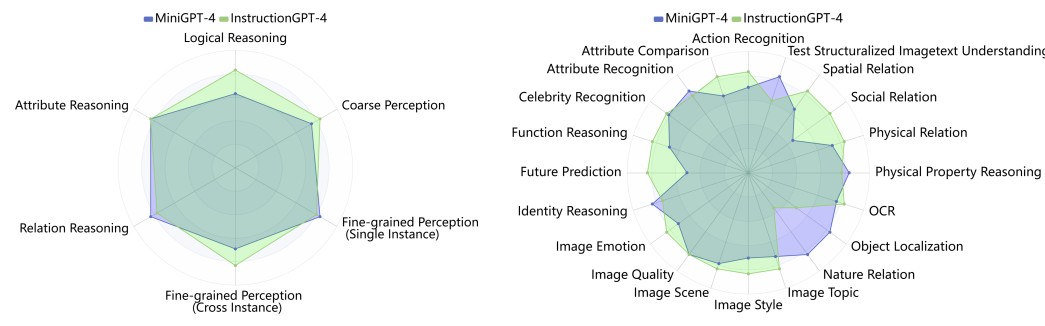

(a) Comparison of 6 dimension levels.    (b) Comparison of 20 ability dimensions.

Figure 3: Comparison of MMBench evaluation (InstructionGPT-4 vs. MiniGPT-4).

## 3.2 EVALUATION

MLLMs are capable of capturing a wide range of multimodal patterns and relationships. Most are evaluated on publicly available datasets or judged by GPT-4 (OpenAI, 2023). Following this trend, we select several popular benchmarks as follows.

We first choose GQA (Hudson & Manning, 2019), IconQA (Lu et al., 2021), ScienceQA (Lu et al., 2022) and OKVQA (Marino et al., 2019) to evaluate the MLLMs tuned from different subsets and treat each metrics as the genuine quality labels mentioned in Section 2.2.

Additionally, we test the zero-shot ability of MLLMs on various VQA datasets, including DocVQA (Mathew et al., 2021), TextVQA (Singh et al., 2019), STVQA (Biten et al., 2019) and VizWiz (Bigham et al., 2010). We also evaluate the vision and language capabilities in complex multimodal tasks of different models on the recently developed benchmarks including MMBench (Liu et al., 2023c) and MME (Fu et al., 2023). Furthermore, we choose GPT-4 (OpenAI, 2023) as a judge to compare the responses from MiniGPT-4 and InstructionGPT-4 given the images and instructions from LLaVA-Bench (Liu et al., 2023b). The score from GPT-4 is measured by comparing two MLLMs' outputs against a reference answer. Detailed description of these evaluation benchmarks are shown in Section A.5.

## 4 EXPERIMENTAL RESULTS

### 4.1 BENCHMARK SCORES

In this section, we conduct quantitative evaluations of InstructionGPT-4 on several datasets using a zero-shot approach. The comparisons of InstructionGPT-4 with the model tuned from 200 random selected samples and MiniGPT-4 are presented in Table 2, Table 3, and Table 4. This assessment offers a valuable perspective on the efficacy of the data selector in enhancing zero-shot performance across a range of tasks.

We observe that InstructionGPT-4 provides the leading performance on average scores in MME (Table 2) and MMBench (Figure 3 and Table 4), and realizes transcendence in all aspects of VQA datasets (Table 3). Specifically, InstructionGPT-4 demonstrates a +23 score improvement over MiniGPT-4 on MME, +1.55 score on MMBench, and +1.76% on VQA datasets. In addition, InstructionGPT-4 outperforms MiniGPT-4 in 8 out of 14 tasks in MME, 4 out of 6 levels as well as 13 out of 20 abilities in MMBench (detailed in Appendix B.1), and all 4 VQA datasets in LVLM-eHub. Moreover, InstructionGPT-4 exceeds the model trained from random samples on all other tasks.

By evaluating and contrasting these models in a range of tasks, we aim to ascertain the efficacy of our proposed data selector that can effectively identify high-quality data. Though our data selector is trained on a list of VQA validation sets, InstructionGPT-4 still demonstrates a strong generation ability to out-domain evaluation datasets such as MME and MMBench. This comprehensive analysis sheds light on the benefits of informed multimodal data selection in enhancing zero-shot performance across diverse and complex tasks.

| Evaluation Tasks | | MiniGPT-4 (3439 samples) | Random Selection (200 samples) | InstructionGPT-4 (200 samples) |
|---|---|---|---|---|
| Perception | Existence | **75.00** | 56.67 ± 10.00 | 73.33 |
| | Count | 30.00 | 23.89 ± 0.96 | **31.67** |
| | Position | 36.67 | 33.33 ± 3.33 | **38.33** |
| | Color | **38.33** | 30.00 ± 13.64 | 36.67 |
| | Poster | 35.71 | 32.99 ± 1.56 | **36.05** |
| | Celebrity | **62.06** | 50.69 ± 1.79 | 59.71 |
| | Scene | 52.75 | 46.00 ± 2.78 | **57.75** |
| | Landmark | 22.25 | 18.92 ± 3.55 | **29.25** |
| | Artwork | 23.50 | 16.91 ± 2.27 | **50.50** |
| | OCR | **57.50** | 55.00 ± 4.33 | 50.00 |
| | Total Score | 433.77 | 364.40 ± 34.92 | **463.26** |
| Cognition | Commonsense Reasoning | 46.43 | 34.52 ± 2.70 | 40.00 |
| | Numerical Calculation | 47.50 | 45.00 ± 2.50 | **50.00** |
| | Text Translation | **50.00** | 44.17 ± 1.44 | 47.50 |
| | Code Reasoning | **47.50** | 40.17 ± 3.82 | **47.50** |
| | Total Score | **191.43** | 162.85 ± 1.29 | 185.00 |
| | Score of MME | 625.20 | 527.26 | **648.26** |

Table 2: Performance comparison on MME.

| Datasets | MiniGPT-4 (3439 samples) | Random Selection (200 samples) | InstructionGPT-4 (200 samples) |
|---|---|---|---|
| STVQA | 13.71 | 13.51 ± 0.73 | **14.55** |
| VizWiz | 46.60 | 44.47 ± 0.86 | **51.02** |
| DocVQA | 2.77 | 2.58 ± 0.23 | **3.01** |
| TextVQA | 19.06 | 18.92 ± 0.31 | **20.62** |
| Average Score | 20.54 | 19.87 | **22.30** |

Table 3: Performance comparison on VQA tasks.

| Dimension Level | MiniGPT-4 (3439 samples) | Random Selection (200 samples) | InstructionGPT-4 (200 samples) |
|---|---|---|---|
| LR | 12.50 | 17.23 ± 1.76 | **16.48** |
| AR | **41.87** | 35.41 ± 5.09 | **41.87** |
| RR | **12.68** | 9.23 ± 2.11 | 11.74 |
| FP-C | 17.60 | 13.20 ± 3.39 | **21.20** |
| FP-S | **35.75** | 26.92 ± 4.14 | 34.25 |
| CP | 38.30 | 29.29 ± 5.55 | **42.55** |
| Score of MMBench | 29.87 | 23.95 | **31.42** |

Table 4: Performance comparison on MMBench.

## 4.2 GPT-4 EVALUATION

Given the presence of inherent position bias within LLMs as evaluators, wherein certain positions are favored over others (Wang et al., 2023), we have undertaken measures to address this concern. To mitigate such bias, we conduct evaluations using both response orders – placing InstructionGPT-4's generated response before and after MiniGPT-4's response. To establish a definitive judgment criterion, we introduce the "Win-Tie-Fail" framework, characterized as follows:

1) Win: InstructionGPT-4 is deemed the winner in two instances, or secures victory once and achieves a draw once; 2) Tie: InstructionGPT-4 achieves a draw twice, or prevails in one instance and succumbs in another; 3) Fail: InstructionGPT-4 faces defeat in two instances, or experiences a loss once and attains a draw once.

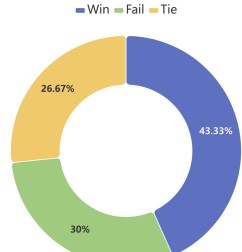

Figure 4: GPT-4 Evaluation Comparison (InstructionGPT-4 vs. MiniGPT-4).

The results of this evaluation are depicted in Figure 4. Win, Fail, and Tie in this figure denote comparative outcomes when the generation results of InstructionGPT-4 are evaluated against those of MiniGPT-4. Throughout 60 questions, InstructionGPT-4 emerges victories in 26 instances, experiences failure in 16, and achieves a tie in 18. This evidence underscores the notable superiority of InstructionGPT-4's response quality in comparison to MiniGPT-4.

## 4.3 DEMONSTRATIONS

We conducted a comparative assessment of image understanding and conversation abilities between InstructionGPT-4 and MiniGPT-4, focusing on a challenging instance described in Table 11 and Table 12. This highlights InstructionGPT-4's exceptional ability for advanced reasoning, emphasizing its prowess in image comprehension and executing instruction-following tasks. The comparison between several selected samples versus original samples is showcased in Appendix C.

## 4.4 ABLATION STUDY

Through a series of ablation studies, we elucidate the contributions of various factors to the overall effectiveness of our data selection approach. Our experimental analyses serve as empirical verification of the theoretical foundations we have put forward.

**Analysis of Different Indicators.** To comprehensively evaluate the impact of distinct indicators on the data selection process, we conduct another ablation study. Each individual indicator is isolated and its effect on 200 data selection is scrutinized. As showcased in the left part of Table 5, the models fine-tuned using data selected based on CLIP Score, Length Score, Reward Score, GPT Score used in Alpagasus (Chen et al., 2023), and Multimodal Features consistently outperform those generated

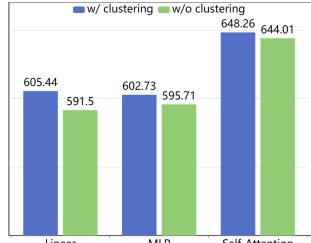 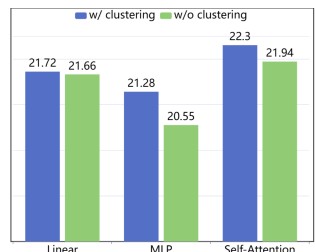

(a) Comparison of MME evaluation.    (b) Comparison of VQA evaluation.

Figure 5: Ablation study to investigate the impact of clustering in the testing stage and different types of network structures utilized in the data selector. Note that self-attention with clustering consistently yields leading performance.

through random sampling. This illustrates that employing each separate indicator yields positive effects on the data selection process, thus they are suitable for data quality assessment.

| Benchmark | Indicators | | | | | | Self-Attention Layers | | |
|---|---|---|---|---|---|---|---|---|---|
| | Random | CLIP | Reward | Length | GPT | Features | 1 | 2 | 3 |
| MME | 527.26 | 95.79 ↑ | 33.36 ↑ | 90.76 ↑ | 59.69 ↑ | 20.82 ↑ | 521.10 | **648.26** | 594.84 |
| VQA | 19.87 | 2.57 ↑ | 1.47 ↑ | 2.39 ↑ | 2.11 ↑ | 0.63 ↑ | 22.08 | **22.30** | 21.74 |

Table 5: MME and VQA scores under different indicators used separately (left) and different self-attention layers (right).

**Analysis of the Data Selector Architecture.** In this ablation study, we try three different structures for data selector, including linear regression used in Instruction Mining (Cao et al., 2023), MLP, and self-attention. Compared to MLP or linear models that can only achieve global awareness, self-attention mechanisms enable internal information interaction within embeddings. The results depicted in Figure 5 indicate the self-attention structure achieves the highest performance. In addition, we conduct experiments with different numbers of attention layers, as summarized in the right part of Table 5. Notably, we find that employing 2 layers is the most suitable configuration.

**Analysis of Clustering.** The application of spectral clustering within the data selector mechanism ensures the diversity of the chosen vision-language instruction data. We conduct an ablation study by removing the clustering mechanism. The results are presented in Figure 5 by comparing the two bars for each data selector structure. Incorporating clustering into data selectors with different structures consistently yields improved performance, highlighting the significance of clustering in enhancing the fine-tuning procedure.

**Analysis of Multimodal Feature Size.** We also explore various sizes of multimodal features after conducting unsupervised dimensionality reduction. The results presented on the left side of Figure 6 demonstrate that setting this size to 6 consistently yields the best performance. Our analysis indicates that when dimensionality reduction is configured with a low value, it may excessively compress multimodal features. Conversely, if dimensionality reduction is set too high, it can lead to an expansion in the embedding's dimensionality, thereby increasing the number of training parameters in data selector. This heightened dimensionality can make the data selector more susceptible to overfitting or underfitting issues.

**Analysis of Selected Data Size.** We aim to identify the minimum amount of data required to make InstructionGPT-4 surpass MiniGPT-4. From the right side of Figure 6, we discover that selecting 50 data points is sufficient when considering the VQA dataset for evaluation. However, due to the gap between the datasets used for evaluation, such as MME and MMBench, 200 data points are needed for fine-tuning the model to comprehensively outperform MiniGPT-4. This observation underscores the strong transferability of our designed data selector. Further analysis is provided in Appendix B.2.

**Analysis of Different Data Pruning Methods.** We conduct additional experiments on two state-of-the-art data pruning methods including EL2N (Paul et al., 2021) and prototypicality (Sorscher et al., 2022). We represent our experiment results in Appendix B.3. We observe that these previous methods can't achieve competitve performance for multimodal LLM compared to our data selector, which showcases that our novel multimodal data selection method is quite necessary.

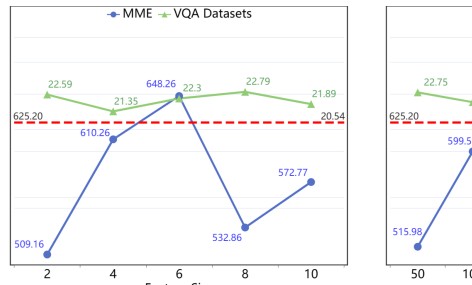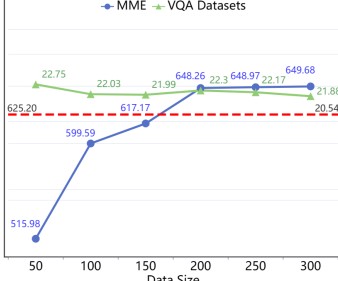

Figure 6: The left part denotes different multimodal feature sizes. The right part denotes different curated data sizes. The red dotted lines represent the performance of MiniGPT-4.

**Analysis of the Stability and Generalizability of Data Selector.** We try multiple random seeds for our data selector's training detailed in Appendix B.4 and achieve consistent superior performance, which shows the stability and robustness of our selection method. We also extend our investigation to include additional multimodal LLMs and vision-language datasets. Based on MiniGPT-4 (Zhu et al., 2023) and Qwen-VL (Bai et al., 2023), we select subsets of the detail_10k (Liu et al., 2023b) dataset without retraining the data selector. By comparing the performance between models tuned from selected subsets and the whole dataset, we observe that less instruction data for better performance still work for various MLLMs. Our experimental results in Appendix B.5 showcase that our data selector is suitable for different types of MLLMs and multimodal datasets.

## 5 RELATED WORKS

**Visual Instruction Tuning.** Instruction tuning is a learning paradigm that fine-tunes pre-trained LLMs on datasets described by natural language instructions, through which the zero-shot abilities of LLMs can be significantly enhanced. The effectiveness of instruction tuning has been demonstrated by a set of research, including FLAN (Wei et al., 2021), InstructGPT (Ouyang et al., 2022), and ChatGPT. Inspired by this, several recent works aim at enabling LLMs to handle multimodal tasks with visual instruction tuning, such as MiniGPT-4 (Zhu et al., 2023), LLaVA (Liu et al., 2023b) and InstructBLIP (Dai et al., 2023). These works choose simple projection layers as the bridges between image encoders and LLMs, and perform visual instruction tuning either on self-instruct datasets (Zhu et al., 2023; Liu et al., 2023b) or on existing multimodal datasets (Zhang et al., 2023a; Dai et al., 2023).

**Instruction Curation.** To improve model performance after instruction tuning, some relevant works manage to filter low-quality instruction data or construct carefully curated examples during the fine-tuning stage, thereby enhancing model capabilities. Previous works (Ghorbani & Zou, 2019; Sehwag et al., 2020; Meding et al., 2021; Sorscher et al., 2022; Paul et al., 2021) have proposed data pruning metrics in vision modal, especially in classification tasks. Meta-Weight-Net (Shu et al., 2019) involves generating weights for selected training samples based on meta learning. Recently, LIMA (Zhou et al., 2023) shows that fine-tuning LLaMA (Touvron et al., 2023b) on 1000 human curated and high-quality examples can produce remarkable results. Several following works (Cao et al., 2023; Chen et al., 2023) have developed instruction quality evaluation methods for measuring the quality of language datasets. Unlike the aforementioned works in a single modal, InstructionGPT-4 is the first multimodal large language model that achieves better performance through effective data selection in generative tasks.

## 6 CONCLUSION

In this paper, we provide a thorough analysis of the data selector's effectiveness in curating valuable multimodal instruction data for generative tasks. We also extensively evaluate InstructionGPT-4's performance on various datasets, confirming its excellence in alignment. InstructionGPT-4's success underscores that inducing instruction data by proper selection can lead to significant advancements for multimodal LLMs, fostering improved instruction understanding and generation capabilities.

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

# Appendix

## A    IMPLEMENTATION DETAILS OF INSTRUCTIONGPT-4

In this section, we first present the design of our GPT-4 prompt for rating scores and the genuine quality labels for training the data selector. We then provide an implementation of our data selection algorithm and further experimental settings.

### A.1    GPT PROMPT

We provide the detailed prompt to GPT-4 used for rating scores in Table 6. It is similar to the prompt for rating and filtering training data in Alpagasus (Chen et al., 2023).

| | GPT Prompt |
|---|---|
| System Prompt | We would like to request your feedback on the performance of an AI assistant. The assistant provides a caption based on an image and an instruction.
Instruction: [Instruction]
Caption: [Caption] |
| User Prompt | Please rate according to the quality and variety of the caption to the instruction. Each assistant receives a score on a scale of 0 to 100, where a higher score indicates higher level of the quality and variety. Please first output a single line containing the value indicating the scores. In the subsequent line, please provide a comprehensive explanation of your evaluation, avoiding any potential bias. The instruction and caption are displayed following without image. |

Table 6: Prompt $p_G$ to GPT-4 for rating multimodal data.

### A.2    GENUINE QUALITY LABELS

To acquire genuine quality labels for the data, we choose to partition the cc_sbu_align dataset into 30 subsets using clustering techniques such as $K$-means++ (Arthur & Vassilvitskii, 2007). Each of these subsets, denoted as $i \in \{1, 2, \ldots, 30\}$, comprises 114 data points. Subsequently, each subset is employed to fine-tune a pre-trained Multimodal Language Model (MLLM). These fine-tuned models are then evaluated on a validation set, including GQA (Hudson & Manning, 2019), IconQA (Lu et al., 2021), OKVQA (Marino et al., 2019) and ScienceQA (Lu et al., 2022), to generate scores in Table 7, which serves as genuine quality labels for the respective subset. We choose these four VQA datasets to produce genuine quality labels because they are sufficiently diverse and contain various question-answer pairs. In particular, GQA focuses on reasoning skills and combined language understanding skills; IconQA requires perceptual skills such as object recognition and text understanding; OKVQA is a large-scale dataset requiring external knowledge; ScienceQA can well diagnose whether the multimodal LLM has multi-step reasoning capabilities and interpretability. These four datasets are wide enough to cover multiple aspects of multimodal tasks.

### A.3    SELECTION ALGORITHM IMPLEMENTATION DETAILS

In the training stage, assume that each data point in our dataset is represented by a $d$-dimensional feature vector. When we form a cluster of $q$ data points, these $d$-dimensional vectors are concatenated to create a feature vector with a shape of $(q, d)$. In training the data selector $F$, we utilize these $(q, d)$ shape vectors. Each cluster, represented by its $(q, d)$ vector, is associated with a genuine quality label that reflects the collective quality of all $q$ data points within the cluster. This approach allows us to train the data selector on a rich and detailed representation of data clusters. Regarding the application of $F$ on a single data point as discussed in Section 2.3, it's important to clarify that $F$ is trained on clusters but is capable of evaluating individual data points as well. When applied to a single data point, $F$ operates on its $(1, d)$ feature vector. This flexibility is a key aspect of $F$, allowing it to function effectively both at the cluster level during training and at the individual data point level during testing.

| Subset | GQA | IconQA | OKVQA | ScienceQA | Average |
|---|---|---|---|---|---|
| 1 | 28.48 | 35.88 | 37.11 | 21.98 | 30.86 |
| 2 | 29.17 | 37.78 | 35.85 | 21.29 | 31.02 |
| 3 | 27.21 | 35.35 | 33.83 | 21.63 | 29.51 |
| 4 | 28.13 | 35.64 | 36.77 | 21.75 | 30.57 |
| 5 | 28.25 | 35.75 | 36.26 | 23.56 | 30.95 |
| 6 | 28.72 | 35.92 | 35.67 | 22.46 | 30.69 |
| 7 | 28.08 | 35.18 | 36.24 | 22.28 | 30.45 |
| 8 | 28.20 | 35.71 | 36.21 | 22.06 | 30.55 |
| 9 | 28.49 | 36.79 | 37.42 | 23.32 | 31.51 |
| 10 | 27.40 | 37.89 | 35.71 | 23.78 | 31.20 |
| 11 | 27.84 | 37.57 | 36.39 | 23.65 | 31.36 |
| 12 | 30.68 | 35.36 | 36.49 | 21.60 | 31.03 |
| 13 | 27.68 | 38.64 | 36.78 | 23.65 | 31.69 |
| 14 | 29.00 | 37.23 | 36.69 | 23.85 | 31.69 |
| 15 | 28.31 | 38.03 | 36.00 | 23.90 | 31.56 |
| 16 | 28.82 | 34.91 | 35.24 | 21.40 | 30.09 |
| 17 | 27.10 | 35.14 | 35.02 | 22.44 | 29.93 |
| 18 | 27.70 | 35.76 | 36.02 | 22.03 | 30.38 |
| 19 | 29.33 | 37.04 | 35.98 | 23.19 | 31.38 |
| 20 | 28.75 | 36.58 | 35.92 | 22.59 | 30.96 |
| 21 | 29.67 | 36.33 | 36.52 | 22.91 | 31.36 |
| 22 | 27.68 | 36.67 | 36.27 | 22.05 | 30.67 |
| 23 | 28.68 | 36.54 | 36.47 | 23.63 | 31.33 |
| 24 | 29.45 | 37.31 | 35.98 | 22.74 | 31.37 |
| 25 | 26.77 | 35.50 | 34.69 | 22.71 | 29.92 |
| 26 | 28.62 | 34.90 | 35.00 | 22.08 | 30.15 |
| 27 | 26.52 | 35.71 | 34.21 | 23.00 | 29.86 |
| 28 | 27.48 | 36.53 | 35.58 | 22.97 | 30.64 |
| 29 | 27.93 | 34.54 | 36.00 | 22.79 | 30.31 |
| 30 | 28.53 | 36.98 | 37.34 | 23.68 | 31.64 |

Table 7: 30 genuine quality labels.

In the testing stage, we proceed to split the multimodal dataset for fine-tuning into $K = 10$ groups for data selection. This division is achieved by employing spectral clustering on the image embeddings, which have been encoded using ViT from CLIP (Radford et al., 2021). The purpose of this clustering step is to ensure diversity within our selected data, as it helps capture a wide range of data distribution patterns. It's important to note that the clustering algorithm used here (denoted as $\Lambda$) is distinct from the one used earlier (denoted as $\Gamma$) for dividing the subsets and obtaining genuine quality labels. Each subset created using $\Gamma$ shares the same genuine quality labels.

Besides, to acquire genuine quality labels (which necessitate training a multimodal model on each cluster group for evaluation) without letting the quantity of data impact the capabilities of the trained multimodal models, we employ a simple post-processing technique for $\Gamma$. This involves identifying clusters with either excessively high or low sample counts, and redistributing some samples from the larger clusters to the smaller ones based on their distance to different cluster centroids, thereby equalizing the number of samples in each cluster.

By introducing this differentiation between $\Lambda$ and $\Gamma$ for data selection after training the data selector, we ensure that predicted labels for data points within each cluster maintain their distinctiveness, preventing potential label confusion. The whole algorithm for testing stage is shown in Algorithm 1.

## A.4 EXPERIMENTAL SETTING DETAILS

When computing the indicators for each triplet containing images, instructions, and responses, we do not take instructions into consideration because they have fixed formats in the dataset (e.g., "Describe this image in detail.").

To generate multimodal features, we employ Principal Component Analysis (PCA) to reduce the dimensionality of the multimodal features generated by frozen ViT (Ilharco et al., 2021) and Llama2 (Touvron et al., 2023b).

---

**Algorithm 1** DATA SELECTION

---

**Require:** Dataset $D$, Trained Data Selector $F$, number of clusters $K$, subset size factor $\alpha$
1: Compute clusters $D_1, D_2, \ldots, D_K$ using a clustering algorithm $\lambda$ on images in $D$
2: **for** $i = 1$ to $K$ **do**
3:     **for** $x$ in $D_i$ **do**
4:         Compute CLIP Score $C(x)$, Length Score $L(x)$, Reward Score $R(x)$, GPT Score $G(x, p_G)$
            and Multimodal Features $P(f(x_{\text{image}}), g(x_{\text{response}}))$
5:         Concat the indicators as embedding $e(x)$ in Equation equation 1
6:         Compute predicted label $F(e(x))$
7:     **end for**
8:     Compute $|S_i| = \frac{\alpha \cdot |D_i|}{|D|}$
9:     Select top $|S_i|$ samples from $D_i$ based on $F(e(x))$ to form $S_i$
10: **end for**
11: Combine $S_1, S_2, \ldots, S_K$ to form $S$
12: **return** $S$

---

For the data selector training, we set the number of training epochs to 20 and the learning rate to 0.01. We conduct all instruction tuning on pre-trained 7B MiniGPT-4 (Zhu et al., 2023) and use the same fine-tuning hyperparameters as the original MiniGPT-4.

For the ablation study of different indicators, we follow the testing stage of the data selector and sort each data point after conducting clustering. We sort each data point based on CLIP Score, Reward Score, and GPT Score respectively. Additionally, we select a length range to evaluate the length effect. Besides, we train another self-attention network with multimodal features as inputs for data selection.

### A.5 EVALUATION BENCHMARKS

**MME** (Fu et al., 2023). It is an MLLM evaluation benchmark that measures both perception and cognition abilities on a total of 14 subtasks. The full score for the overall tasks is 2800, while for the subtasks is 200. For each test image, MME adopts an instruction of a question and a description "Please answer yes or no" to prompt MLLMs. Such a concise instruction-answer evaluation allows for a fair comparison of MLLMs without the impact of prompt engineering.

**MMBench** (Liu et al., 2023c). This benchmark is collected from multiple sources, including public datasets and Internet, and currently, contains 2974 multiple-choice questions, covering 20 ability dimensions. The existing 20 ability dimensions are structured into 6 dimension levels. Each question is a multiple-choice format with a single correct answer. For a more reliable evaluation, it employs ChatGPT to match a model's prediction with the choices of a question, and then output the corresponding label (A, B, C, D) as the final prediction.

**VQA Datasets**. LVLM-eHub (Xu et al., 2023) is a comprehensive evaluation benchmark for publicly available MLLMs. Based on this platform, we choose GQA (Hudson & Manning, 2019), IconQA (Lu et al., 2021), ScienceQA (Lu et al., 2022) and OKVQA (Marino et al., 2019) to evaluate the MLLMs tuned from different subsets and treat each metrics as the genuine quality labels mentioned in Section 2.2. We also test the zero-shot ability of MLLMs on various datasets, including DocVQA (Mathew et al., 2021), TextVQA (Singh et al., 2019), STVQA (Biten et al., 2019) and VizWiz (Bigham et al., 2010). Top-1 accuracy is employed for these tasks.

**LLaVA-Bench** (Liu et al., 2023b). It collects a diverse set of 24 images with 60 questions in total, including indoor and outdoor scenes, memes, paintings, sketches, etc. It associates each image with a highly detailed and manually curated description and a proper selection of questions that are categorized into conversation (simple QA), detailed description, and complex reasoning. We choose GPT-4 as a judge to compare the responses from MiniGPT-4 and InstructionGPT-4 given the images and instructions from LLaVA-Bench. The score from GPT-4 is measured by comparing two MLLMs' outputs against a reference answer. Such a design assesses the model's robustness to different prompts.

# B MORE EXPERIMENTAL RESULTS

## B.1 MMBENCH RESULTS

MMBench (Liu et al., 2023c) gathers approximately 3000 questions spanning 20 ability dimensions in 6 levels. The detailed assessment of 20 abilities is illustrated in Table 8.

| Ability Dimension | MiniGPT-4 (3439 samples) | Random Selection (200 samples) | InstructionGPT-4 (200 samples) |
|---|---|---|---|
| Action Recognition | 37.50 | 28.03 ± 6.84 | **44.32** |
| Attribute Comparison | 5.00 | 3.75 ± 2.04 | **6.25** |
| Attribute Recognition | **51.00** | 41.33 ± 2.87 | 48.00 |
| Celebrity Recognition | 34.75 | 22.03 ± 7.22 | **35.59** |
| Function Reasoning | 34.58 | 29.91 ± 6.05 | **42.06** |
| Future Prediction | 18.92 | **34.23 ± 2.78** | 31.08 |
| Identity Reasoning | **69.51** | 56.91 ± 6.40 | 62.20 |
| Image Emotion | 35.71 | 38.89 ± 1.48 | **41.67** |
| Image Quality | **3.49** | 1.94 ± 1.45 | **3.49** |
| Image Scene | 66.15 | 49.74 ± 11.17 | **70.00** |
| Image Style | 31.76 | 19.22 ± 9.72 | **37.65** |
| Image Topic | 40.00 | 26.27 ± 4.93 | **45.88** |
| Nature Relation | **17.28** | 6.17 ± 1.01 | 7.41 |
| Object Localization | **16.19** | 12.38 ± 0.78 | 9.52 |
| OCR | 44.16 | 35.50 ± 6.12 | **48.05** |
| Physical Property Reasoning | **27.00** | 23.67 ± 3.09 | 25.00 |
| Physical Relation | 13.46 | 14.10 ± 3.27 | **15.38** |
| Social Relation | 7.50 | 9.17 ± 4.13 | **13.75** |
| Spatial Relation | 8.54 | 6.50 ± 1.52 | **10.98** |
| Test Structuralized Imagetext Understanding | **7.84** | 4.90 ± 1.60 | 5.88 |
| Score of MMBench | 29.87 | 23.95 | **31.42** |

Table 8: Performance comparison on MMBench.

## B.2 FURTHER ANALYSIS IN SELECTED DATA SIZE

Our findings reveal that our data selection method outperforms random selection in all cases except the size of 1000 in the right side of Figure 7. Through our observation, there is an interesting trend as the size of the selected subset increases: the performance gap between our method and random selection begins to narrow.

This decrease in the performance differential can be attributed to the nature of our data selection process. As the size of the subset selected by our method increases, it inevitably starts to include more data points of lower quality. This inclusion of lower-quality data diminishes the overall effectiveness of the selected subset, thereby reducing the gap in performance compared to a randomly selected subset.

Despite this trend, it is important to note that our data selection method continually maintains a performance advantage over random selection in most cases. This finding underscores the efficacy of our approach, particularly when working with smaller subsets. It also highlights a key insight: the quality of data, not just the quantity, is crucial for improving model performance.

## B.3 COMPARISON OF DIFFERENT DATA PRUNING METHODS

Since the evaluation tasks have different data distributions and different task types from the training task (e.g., the evaluation task is yes/no question, while the selector training task is general VQA), the smaller training loss does not directly indicate better performance on the evaluation task. However, previous data pruning method mainly rely on the model loss from the training dataset, which would be unfair to them if they were used directly. Alternatively, a small number of data pruning methods requires access to the evaluation metric, which involve such extensive computations that they are impractical for use with large models. Thus, we design a learning-based method by formulating a

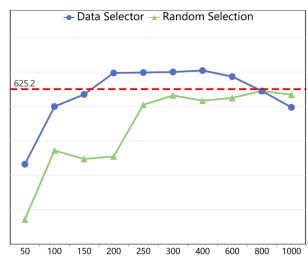 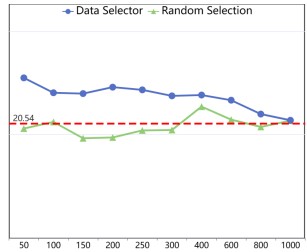

(a) Comparison of MME evaluation. (b) Comparison of VQA evaluation.

Figure 7: Comparison between data selector and random selection over different data sizes.

set of indicators for assessment and train a neural network as a data selector to fit the indicators to the genuine quality labels for selection.

Moreover, previous data pruning methods have several important differences from our proposed method. While data pruning is a model performance driven method, we adopt a learning-based method by training a data selector, which can be applied to new datasets independent of certain models. Besides, most of data pruning methods focus on optimizing models based on the loss, while we extract the multimodal data indicators to train the selector based on MLLM performance, which can reduce the gap of data distribution in evaluation set. Furthermore, data pruning aims at cutting off bad data for training, while we concentrate on data mining for selecting a subset with least and best data.

To demonstrate the superiority of our data selection method, we also compare our proposed selector with two state-of-the-art data pruning methods including EL2N (Paul et al., 2021) and prototypicality (Sorscher et al., 2022). In detail for EL2N, we compute for every training example in cc_sbu_align dataset the average L2 norm of the error vector (EL2N score). We conduct data pruning by retaining only the 200 hardest examples with largest error on the pre-trained model. As for prototypicality, we perform k-means clustering in the embedding space of a pre-trained model (here: ViT (Ilharco et al., 2021)), and select 200 hardest data based on the difficulty of each data point by the Euclidean distance to its nearest cluster centroid, or prototype. The reults in the left side of Table 9 showcases that data selected by our multimodal selector works much better than these two methods.

### B.4 RESULTS OVER MULTIPLE RANDOM SEEDS WITH REGARDS TO THE DATA SELECTION

We conduct additional experiments to try different random seeds. These experiments help us evaluate the stability and generalizability of our selection method under different initialization conditions. According to the right part of Table 9, we observe that the performance variance across different seeds is not obvious. This consistency of the performance underscores the effectiveness of our data selection method, suggesting that it is not overly sensitive to the initial random seed and can reliably identify high-quality data subsets across different scenarios.

| Benchmark | Different Data Pruning Methods | | | Multiple Random Seeds | | | | |
|---|---|---|---|---|---|---|---|---|
| | Random | EL2N | Prototypicality | Seed 1 | Seed 2 | Seed 3 | Seed 4 | Average |
| MME | 527.26 | 627.27 | 569.46 | 648.26 | 637.58 | 628.99 | 649.25 | $641.02 \pm 8.32$ |
| VQA | 19.87 | 20.62 | 21.83 | 22.30 | 21.78 | 22.14 | 21.98 | $22.05 \pm 0.19$ |

Table 9: MME and VQA scores under different data pruning methods (left) and multiple random seeds (right).

### B.5 SCALABILITY AND GENERALIZABILITY OF THE DATA SELECTOR

We apply our data selection method to Qwen-VL (Bai et al., 2023) and the detail_10k (Liu et al., 2023b) dataset without retraining the data selector. We select different sizes of subsets, including 1K and 2K, from the detail_10k dataset and compare their performance of the models tuned from them.

Qwen-VL is a strong pre-trained multimodal model uses Qwen-7B as the initialization of the LLM, and ViT as the initialization of the visual encoder. It connects them with a randomly initialized

cross-attention layer, which is trained with alignment techniques. Qwen-VL supports more flexible interaction, such as multiple image inputs, multi-round question answering, and creative capabilities. Detail_10k dataset includes a rich and comprehensive description for each image. The description is created by prompting GPT-4.

Given the detail_10k dataset, we extract the indicators of each data point to feed into the trained data selector for selection. Our findings in Table 10 indicate promising results, suggesting that multimodal instruction data curated by our data selector directly works well on different MLLMs. Thus, our method is not only effective with MiniGPT-4 and cc_sbu_align dataset but also exhibits potential for broader applicability.

| Detail_10K Dataset | Qwen-VL | | | MiniGPT-4 | | |
|---|---|---|---|---|---|---|
| | 1K | 2K | 10K | 1K | 2K | 10K |
| MME | 1802.50 | **1806.81** | 1769.28 | **614.04** | 608.54 | 604.47 |
| Perception | 1423.57 | **1426.10** | 1398.92 | **434.04** | 431.04 | 430.74 |
| Cognition | 378.93 | **380.71** | 370.36 | **180.00** | 177.50 | 173.93 |

Table 10: MME (including Perception and Cognition) scores under different types of Multimodal LLMs and different sizes of selected data from detail_10k dataset.

### B.6 ADDITIONAL DEMONSTRATIONS

We display both InstructionGPT-4 and MiniGPT-4 multimodal chatbot demos below.

Table 11 and Table 12 distinctly indicate that InstructionGPT-4 possesses the capacity to generate responses that are not only more comprehensive but also exhibit a higher level of fluency when compared to those produced by MiniGPT-4.

The story presented by InstructionGPT-4 in Table 11 is vivid and detailed, which involves the key information from the image. Furthermore, when tasked with writing a recipe based solely on a food image in Table 12, InstructionGPT-4 demonstrates a notably superior ability in chain-of-thought by generating robust and effective responses step by step. These findings collectively emphasize the enhanced language generation prowess of InstructionGPT-4 as it is trained on higher-quality instruction dataset with better image-text alignment and more fluent captions, which contributes to generate more detailed, expressive, and contextually appropriate responses in various scenarios.

## C SELECTED SAMPLES VERSUS ORIGINAL SAMPLES

We compare the selected 200 data points with the original dataset and find that the selected 200 data points exhibit higher quality. Since we extract the image-text pair's multiple indicators for selection, the selected data have higher quality for MLLM's fine-tuning. This is reflected by Figure 8 in the more comprehensive, detailed, and error-free language representation of the selected textual data, with no apparent grammatical or syntactical errors. The data with short, incomplete and inaccurate captions has been filtered out since we apply length, GPT Score and Reward Score as indicators for data selection. Besides, the alignment degree between images and responses becomes better as we include CLIP Score into the indicators. It demonstrates that the data selector can effectively choose data aligned with human preferences.

## D LIMITATION AND FUTURE WORKS

While achieving promising performance, the proposed framework also has some inherent limitations. Developing a more generalized version of the data selector, such as considering more indicators for evaluation, is a potential avenue for exploration. Additionally, undertaking multimodal instruction mining may provide further insights and unveil new opportunities in this domain. This could enable a broader and more comprehensive understanding of this field.

| | Demos |
|---|---|
| | 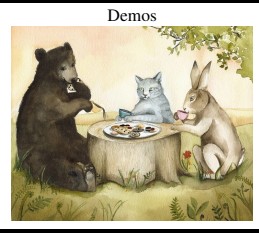 |
| User | Can you write a story around this image for my child? |
| InstructionGPT-4 | Yes, I can write a story around this image.
Once upon a time, there was a group of animals who lived in a forest together. The animals were a bear, a rabbit, and a cat. They all loved to sit down to a meal together, especially for breakfast.
One morning, the bear woke up early and decided to make breakfast for his friends. He went to the stream and caught some fish, then returned to the table to cook them in a pan with butter. The rabbit helped him by bringing him some eggs, and the cat helped by bringing him some cream.
They all sat down at the table under the shade of a tree and enjoyed their breakfast together. The bear said, "This is the best breakfast I've ever had!" The rabbit said, "I'm glad you like it, Bear. This is my favorite too." The cat said, "Meow, meow! This is delicious."
The animals enjoyed their meal and their friendship. They promised to have breakfast together every morning for as long as they lived in the forest. And so they did, until the day they all had to say goodbye to each other, as they each had to go their own way. But the memories of their breakfasts together remained, and they were always happy to think back on the good times they had in the forest. |
| MiniGPT-4 | Once upon a time, there was a bear, a rabbit, and a cat who lived in the forest. They spent their days exploring the forest, playing games, and having picnics together. One day, they decided to have a picnic at a table in the middle of the forest. They sat around the table, enjoying their meal and the beautiful scenery around them. The bear ate his share of the food, the rabbit ate her share, and the cat drank some milk. They were all happy and content. |

Table 11: An Story Generation Example.

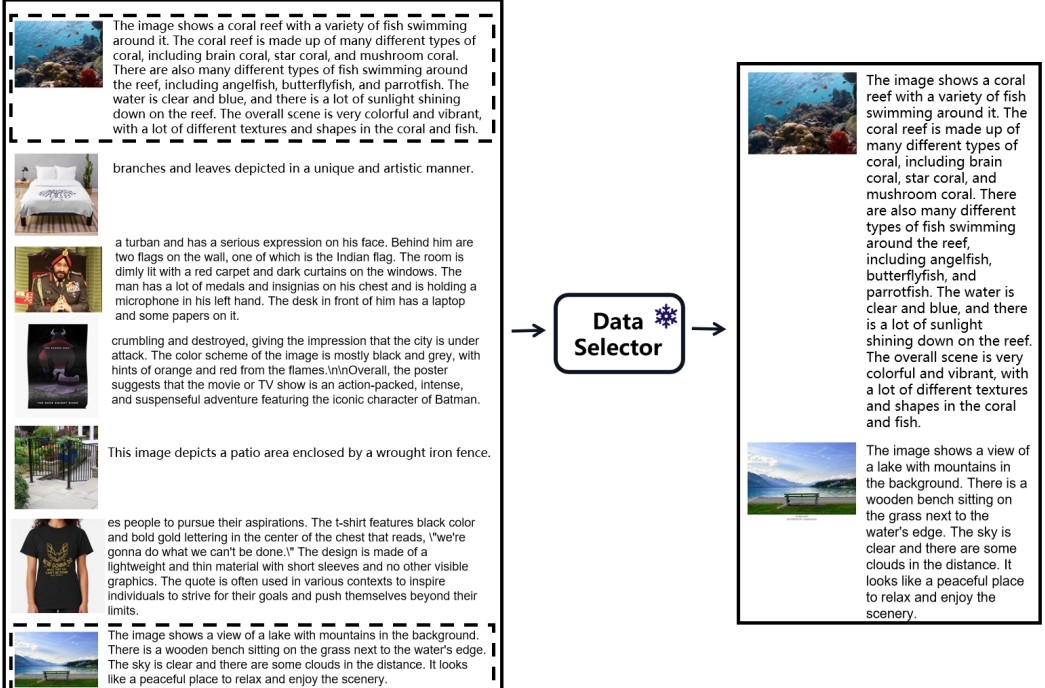

Figure 8: Data selector can filter out low-quality data (e.g., inappropriate grammar and incomplete expressions).

| | Demos |
|---|---|
| | 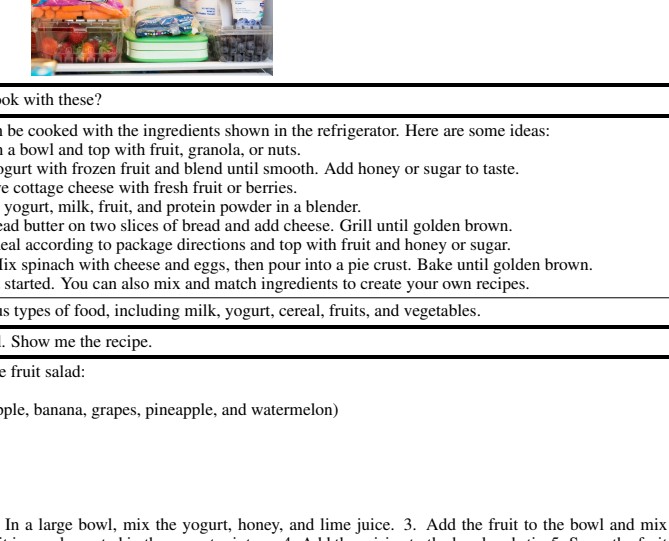 |
| User | What are the meals that I can cook with these? |
| InstructionGPT-4 | There are various meals that can be cooked with the ingredients shown in the refrigerator. Here are some ideas:
* Yogurt parfait: Place yogurt in a bowl and top with fruit, granola, or nuts.
* Fruity smoothies: Combine yogurt with frozen fruit and blend until smooth. Add honey or sugar to taste.
* Cottage cheese and fruit: Serve cottage cheese with fresh fruit or berries.
* Breakfast smoothie: Combine yogurt, milk, fruit, and protein powder in a blender.
* Grilled cheese sandwich: Spread butter on two slices of bread and add cheese. Grill until golden brown.
* Oatmeal and fruit: Cook oatmeal according to package directions and top with fruit and honey or sugar.
* Spinach and cheese quiche: Mix spinach with cheese and eggs, then pour into a pie crust. Bake until golden brown.
These are just a few ideas to get started. You can also mix and match ingredients to create your own recipes. |
| MiniGPT-4 | This refrigerator contains various types of food, including milk, yogurt, cereal, fruits, and vegetables. |
| User | I would like to do the fruit salad. Show me the recipe. |
| InstructionGPT-4 | Sure, here's a recipe for a simple fruit salad:
Ingredients:
* 1 kg of mixed fruit (such as apple, banana, grapes, pineapple, and watermelon)
* 1 cup of yogurt
* 1 tablespoon of honey
* 1 tablespoon of lime juice
* 1/4 cup of raisins
Instructions:
1. Wash and slice the fruit. 2. In a large bowl, mix the yogurt, honey, and lime juice. 3. Add the fruit to the bowl and mix everything together until the fruit is evenly coated in the yogurt mixture. 4. Add the raisins to the bowl and stir. 5. Serve the fruit salad chilled.
You can adjust the amount of fruit and yogurt to your liking. You can also add other ingredients such as granola, nuts, or even chia seeds for added texture and nutrition. Enjoy! |
| MiniGPT-4 | Sure, here's a recipe for a simple fruit salad:
Ingredients:
* Bananas, peeled and sliced
* Apples, peeled and sliced
* Grapes
* Strawberries
* Blueberries
* Rice
* Sugar
* Lemon juice
Instructions:
1. In a large bowl, mix together the sliced bananas, apples, grapes, strawberries, blueberries, and rice. 2. In a separate bowl, mix together the sugar and lemon juice until the sugar is fully dissolved. 3. Pour the sugar mixture over the fruit mixture and toss to coat everything evenly. 4. Serve the fruit salad cold or chilled, garnished with lemon wedges or mint leaves if desired.
That's it! This simple fruit salad is a great way to get in some healthy fruits and veggies while satisfying your sweet tooth. Enjoy! |

Table 12: Generating chain-of-thought responses.

