# OpenReview forum: "InstructionGPT-4: A 200-Instruction Paradigm for Fine-Tuning MiniGPT-4"
_ICLR.cc/2024/Conference — Submitted to ICLR 2024_

### Official Review · Reviewer_PGky · 2023-10-28

**Soundness:** 3 good
**Presentation:** 3 good
**Contribution:** 3 good
**Rating:** 5
**Confidence:** 3

**Summary:**

This paper introduces InstructionGPT-4 which achieves better output with only 200 high-quality examples. The authors propose metrics and a trainable data selector to filter low-quality vision-language data. InstructionGPT-4 outperforms MiniGPT-4, showcasing the efficiency of using less but high-quality instruction data for improving multimodal language models.

**Strengths:**

1. The question posed by the author is highly significant, namely the impact of instruction data quality on the performance of MLLM, which has been rarely discussed before.
2. The method described in the article is written quite clearly and is easy to understand. The entire method is also concise, and the results on several benchmarks demonstrate its effectiveness.

**Weaknesses:**

1. The experiments in the article are relatively limited and the dataset used has a small sample size, which raises doubts about the effectiveness of the proposed method when applied to larger datasets.
2. The authors should have included some baseline methods for data selection, such as the mentioned instruction mining. It is not surprising that removing low-quality data can improve performance, and having these baselines would provide a better comparison and evaluation of the proposed approach.

**Questions:**

1. Regarding the data selector, although it appears to be concise, it can be time-consuming as it requires continuous training of the model on the validation dataset for testing. This becomes particularly challenging when dealing with large amounts of data.
2. The article suggests that the data selector can be directly applied to other multimodality datasets. It would be beneficial if the authors could provide evidence or supporting experiments.
3. The dataset size of 3.4k in MiniGPT-4 is relatively small in terms of instruction data. It would be interesting to investigate whether the proposed method remains effective on larger instruction datasets.

---

> ### Author Response · Authors · 2023-11-18
> **Response to Reviewer PGky (1/2)**
>
> Q1.1: The experiments in the article are relatively limited and the dataset used has a small sample size, which raises doubts about the effectiveness of the proposed method when applied to larger datasets.
>
> Q1.2: The article suggests that the data selector can be directly applied to other multimodality datasets. It would be beneficial if the authors could provide evidence or supporting experiments.
>
> Q1.3: The dataset size of 3.4k in MiniGPT-4 is relatively small in terms of instruction data. It would be interesting to investigate whether the proposed method remains effective on larger instruction datasets.
>
> A1 for Q1.1, Q1.2 and Q1.3: Thanks for your questions. In particular, we want to find out **if there exists a subset making InstructionGPT-4 achieve better performance**. As a result, our proposed method indeed shows the existence of the subset. Using only 6% of the data, we manage to make InstructionGPT-4 surpass MiniGPT-4 in most tasks, which is quite surprising.
>
> To address your concern about scalability and generalizability, we also extend our investigation to include additional models and datasets. **Notably, we have applied our method to Qwen_VL [1] and the detail_10k [2] dataset without retraining the data selector**.
> Here are our experimental results:
>
> | detail_10k   | Qwen_VL (1k) | Qwen_VL (2k) | Qwen_VL (10k) | MiniGPT-4 (1k) | MiniGPT-4 (2k) | MiniGPT-4 (10k) |
> |:------------:|:------------:|:------------:|:-------------:|:--------------:|:--------------:|:---------------:|
> | MME          |    1802.50   |   **1806.81**|    1769.28    |   **614.04**   |      608.54    |      604.47     |
> | Perception   |    1423.57   |   **1426.10**|    1398.92    |   **434.04**   |      431.04    |      428.54     |
> | Cognition    |    378.93    |   **380.71** |    370.36     |   **180.00**    |      177.50    |      173.93     |
>
> By comparing the performance between models tuned from selected subsets and the whole dataset, we observe that less instruction data for better performance still work for various MLLMs. Our findings indicate promising results, suggesting that our method is not only effective with MiniGPT-4's datasets but also exhibits potential for broader applicability.
>
> [1] Bai, et al. Qwen-vl: A frontier large vision-language model with versatile abilities. arXiv preprint arXiv:2308.12966, 2023.
>
> [2] Liu, Li, et al. Visual instruction tuning. arXiv preprint arXiv:2304.08485, 2023.
>
> Q2: The authors should have included some baseline methods for data selection, such as the mentioned instruction mining.
>
> A2: Thanks for your comment. We would like to highlight that **we are the first to propose a data selection method working on multimodal LLM for generative tasks**. Though there are no other data seletion methods for multimodal LLMs, we actually compare our method with single-modal baselines in our ablation study. In particular, **we compare our proposed selector with the recent methods such as Alpagasus (GPT Score only) in Table 5 and Instruction Mining (Linear Selector) in Figure 5**, demonstating that data selected by our multimodal selector works much better than these two methods.
> Moreover, **we also conduct additional experiments on other data pruning methods proposed in single vision modal, including EL2N [1] and prototypicality [2]**. We represent our experiment results below:
>
> | Benchmark | InstructionGPT-4 |  EL2N  | Prototypicality | Random |
> |:---------:|:----------------:|:------:|:---------------:|:------:|
> | MME       |      **648.26**      | 627.27 |     569.46      | 527.26 |
> | VQA       |      **22.30**       |  20.62 |     21.83       |  19.87 |
>
> We observe that these previous methods can't achieve competitve performance for multimodal LLM compared to our data selector. It showcases that our novel data selection method for multimodal datasets is quite necessary. We also add these experiments to the revised version along with the above detailed discussion.
>
> [1] Paul, Ganguli, Dziugaite G K. Deep learning on a data diet: Finding important examples early in training. NeurIPS, 2021.
>
> [2] Sorscher, Ben, et al. Beyond neural scaling laws: beating power law scaling via data pruning. NeurIPS, 2022.

---

> ### Author Response · Authors · 2023-11-18
> **Response to Reviewer PGky (2/2)**
>
> Q3: Regarding the data selector, although it appears to be concise, it can be time-consuming as it requires continuous training of the model on the validation dataset for testing. This becomes particularly challenging when dealing with large amounts of data.
>
> A3: Thank you for your comment. In fact, we have designed our method with efficiency in mind, balancing the need for accurate data selection against the constraints of resource usage.
>
> First of all, based on our data selection method mentioned in Section 2.3, **the time and resource can be lowered by reducing the number of divided subsets, which means training fewer models to gain quality labels**.
>
> Secondly, **the cost of our data selector's training is acceptable and reasonable compared to constructing new high-quality datasets**. Inspired by LIMA [1]'s approach of manually curating 1000 high-quality instruction data for fine-tuning, we take this concept further by **selecting only 200 high-quality multimodal data points from raw datasets**. Our data selection method avoid the need of curating high-quality instruction data from scratch, but select high-quality data from exisiting datasets. This finding is particularly significant given the complexity and resource demands of constructing new datasets.
>
> Most importantly, once our data selector is trained on one dataset, **it can be applied to other datasets without the need for retraining** as mentioned in A1 above. This reusability makes the initial investment in training the selector more cost-effective over time. When dealing the large amounts of data, the data selector can directly identify high-quality data points based on the criteria learned during its initial training.
>
> Thus, our data selector is practical and resource-efficient when dealing with new datasets.
>
> [1] Zhou, Liu, et al. Lima: Less is more for alignment. arXiv preprint arXiv:2305.11206, 2023.

---

> > ### Comment · Reviewer_PGky · 2023-11-22
> > **Response for the rebuttal**
> >
> > Thank you for your detailed rebuttal. Most of the initial concerns have been addressed satisfactorily. The results obtained on the new dataset without retraining are promising.
> >
> > However, I share the same concern as reviewer dCjW regarding the inclusion of a large number of experiments within a short time frame. Other reviewers have also expressed similar concerns, focusing on the simplicity of the chosen baselines and the practicality and generalizability of the proposed method. These concerns should be addressed in the initial version of the paper as they are so crucial for this study.
> >
> > I also suggest that the authors provide more specific examples that are filtered by the algorithm, and more dialogue examples to show what aspect of the model has been enhanced for each dataset. This will enable readers to better comprehend the effectiveness of your method.
> >
> >
> > I appreciate the clarification you have provided and the additional experiments conducted during the rebuttal period. I believe they will greatly enhance the impact of the paper. However, at this time, I have chosen to maintain my original score. I do not deny the importance of your research. It's just that, at present, it doesn't seem to be ready yet.

---

> > > ### Author Response · Authors · 2023-11-23
> > > **Seeking Your Input on Revised Paper's Alignment with ICLR Standards**
> > >
> > > Dear Reviewer PGky,
> > >
> > > We would like to bring to your attention that the discussion period is concluding shortly. We are grateful that the majority of your initial concerns have been successfully addressed and you acknowledge the promise of our results on the new dataset. At the same time, we notice that you still have some unresolved queries. In our most recent response, we made efforts to thoroughly address these remaining questions. Thus, we want to **ensure that we have thoroughly addressed all your concerns and our revised paper fully meets the standards of ICLR**. We would highly value any additional feedback you may provide.
> > >
> > > Thank you sincerely for your time and consideration.
> > >
> > > Best regards,
> > >
> > > The Authors

---

> ### Author Response · Authors · 2023-11-20
> **We would be grateful if you could take a look at the response**
>
> Dear Reviewer PGky:
>
> We sincerely appreciate your valuable time devoted to reviewing our manuscript. We would like to gently remind you of the **approaching deadline for the discussion phase**. We have diligently addressed the issues you raised in your feedback, providing detailed explinations. For instance, we have straightforwardly applied our data selector for different multimodal LLMs and datasets, which achieve promising performance that demonstrate the generalizability of our data selector. Moreover, we have included more baselines by conducting experiments on several existing data pruning methods, which indicate the superiority our multimodal data selection approach. Besides, we clarify that our data selector is practical and resource-efficient when dealing with new datasets. Would you kindly take a moment to look at it?
>
> We are very enthusiastic about engaging in more in-depth discussions with you.

---

> ### Author Response · Authors · 2023-11-22
> **Response to Reviewer PGky**
>
> Dear Reviewer PGky,
>
> Thank you for reviewing our rebuttal and provide further comments. We are happy that most of the initial concerns have been addressed satisfactorily.
>
> In response to the concern about the large number of experiments conducted within a short timeframe, **we have added comprehensive details about the experiments in the revised manuscript**. This includes the rationale behind the selection of experiments, the methodologies employed, and the significance of their outcomes. This additional information aims to alleviate concerns regarding the scope and pace of our experimental work.
>
> Regarding your concern about the simplicity of the chosen baselines and the practicality and generalizability of the proposed method, we have clarified this in our latest version. Specifically, previous data pruning methods as baselines data pruning methods either rely on the model "loss" from the "training dataset" where there exists a gap to the "performance" on the "test dataset" in our problem, or require accesses to the evaluation metric which involve extensive computations and are impractical for large models. In contrast, we adopt a learning-based method by training a data selector, which can be applied to new datasets independent of certain models. Specifically, we extract the multimodal data indicators to train the selector based on MLLM performance, which can reduce the gap of data distribution in evaluation set. Furthermore, data pruning aims at cutting off bad data for training, while we concentrate on identifying a minimal yet effective subset of data, offering a novel perspective in MLLM's training.
>
> We would like to emphasize that reviewer dCjW's concerns regarding the inclusion of a large number of data pruning methods as baselines come from their misunderstanding of our motivation. In fact, **it is unfair for data pruning methods to compare with our method** in this scenario (as they indeed empirically underperform our method). Instead, **we design a learning-based method** by formulating a set of indicators for assessment and train a neural network as a data selector to fit the indicators to the genuine quality labels for selection. Thus, our related baselines should be Instruction Mining [1] and Alpagasus [2], which are included in our initial version of paper.
>
> In addition, **the added experiments on Qwen-VL and detail_10K dataset have demonstrated the practicality and generalizability of the proposed method**. Given specific dataset, we only need to extract their indicators for the trained data selector without conducting any training or inferencing on other MLLMs. The promising results indeed demonstrate the practicality and generalizability of the proposed method compared with other data pruning methods.
>
> To enhance the reader's understanding of our method's impact, **we have incorporated a selection of filtered examples and dialogues in the latest version in Appendix C**. These examples should illustrate how the data selector improves the data quality, providing a clearer understanding of the data selector's impact and effectiveness. Besides, the chatbot demos in Appendix B.6 show how our method enhances different aspects of the model in generation.
>
> Given that most of your concerns have been addressed satisfactorily and we provided further experiments to respond to your remaining questions, we would like to humbly request a reconsideration of the scoring based on our latest version. We believe our research can provide valuable insights and contributions that would be of great interest to the ICLR community.
>
> [1] Cao, Kang, Sun. Instruction mining: High-quality instruction data selection for large language models. arXiv preprint arXiv:2307.06290, 2023.
>
> [2] Chen, Li, et al. Alpagasus: Training a better alpaca with fewer data. arXiv preprint arXiv:2307.08701, 2023.

---

### Official Review · Reviewer_61XF · 2023-10-29

**Soundness:** 2 fair
**Presentation:** 2 fair
**Contribution:** 2 fair
**Rating:** 3
**Confidence:** 4

**Summary:**

This paper proposes a data selection method for finetuning Multimodal Large Language Models (MLLMs), specifically for MiniGPT4 using the training dataset "cc_sbu_align". The authors evaluate the data quality by dividing the whole dataset into some subsets and evaluating the model performance on validation sets. A data selector is trained based on the obtained data quality as labels. The data selector is used to choose a small subset from the whole dataset for final MLLM finetuning. Specifically, 200 data samples are chosen from around 3k samples, and the resulting MLLM has comparable performance as the model finetuned with the whole dataset.

**Strengths:**

The main strength is that the authors managed to train an MLLM with fewer samples and obtained a comparable performance as using the whole dataset.

**Weaknesses:**

- The authors did not demonstrate the generality of the method. There are many nuances in the method and the experiments are done on only one dataset "cc_sbu_align". It is unclear whether the method will work on another dataset or another MLLM.
- There are many systematic methods in the literature that do something similar to select a subset from a dataset based on the model performance on a validation set. A notable one is Meta-Weight-Net [1] based on meta learning. How does this InstructionGPT-4 compare with the Meta-Weight-Net?
- The method obtained slightly better scores than the MiniGPT4 for some scores, but it is partially weaker than MiniGPT4. Moreover, in the benchmarks MME and MMBench, MiniGPT4 is a weak baseline. The marginal improvement over MiniGPT4 is not convincing.
- If the data selector $F$ takes a data sample as input (see questions below for this), then in the paper the authors actually assign the same label to every data sample in the same subset $D_i$, which does not make sense because the label is due to the contribution of the whole subset (the model performance after being trained on the subset).

[1] Shu et al., Meta-Weight-Net: Learning an Explicit Mapping For Sample Weighting

**Questions:**

- What features of the data samples did the authors use for the clustering algorithms? Are the features flattened images or some extracted image features?
- The description in Section 2.3 is confusing. In the "Training" paragraph, it seems that the data selector $F$ maps from the feature of a __whole__ subset to a score, because the index $i$ is for the subsets and it seems to mean $y_i = F(e_i)$. But in the "Testing" paragraph, it seems that the data selector $F$ maps from the feature of a single data sample to a score. Can the authors clarify it?
- Why did the authors not simply use the quality scores to select a subset for finetuning the final MLLM? It seems that the training data and the testing data for the data selector are the same. What is the meaning of performing the testing if we already have the labels on the data?
- What is the relationship between the validation data used to generate the quality labels and the evaluation data? Why could the model performance on validation data be used as an indicator for model performance on the evaluation (test) data?

---

> ### Author Response · Authors · 2023-11-18
> **Response to Reviewer 61XF (1/3)**
>
> Q1: The authors did not demonstrate the generality of the method.
>
> A1: Thank you for your comments. Following LIMA [1] that proposes "less is more for alignment" only using LLaMA in textual modal, our main motivation is to prove that less but high-quality instruction data can outperform the whole dataset for multimodal LLMs. In particular, we want to find out **if there exists a subset making InstructionGPT-4 achieve better performance**. As a result, our proposed method indeed shows the existence of the subset. Besides, we want to focus on MiniGPT-4 because it's the first and most popular multimodal model utilizing LLM.
> To address your concern about scalability and generalizability, we also extend our investigation to include additional models and datasets. **Notably, we have applied our method to Qwen_VL [2] and the detail_10k [3] dataset without retraining the data selector**.
> Here are our experimental results:
>
> | detail_10k   | Qwen_VL (1k) | Qwen_VL (2k) | Qwen_VL (10k) | MiniGPT-4 (1k) | MiniGPT-4 (2k) | MiniGPT-4 (10k) |
> |:------------:|:------------:|:------------:|:-------------:|:--------------:|:--------------:|:---------------:|
> | MME          |    1802.50   |   **1806.81**|    1769.28    |   **614.04**   |      608.54    |      604.47     |
> | Perception   |    1423.57   |   **1426.10**|    1398.92    |   **434.04**   |      431.04    |      428.54     |
> | Cognition    |    378.93    |   **380.71** |    370.36     |   **180.00**    |      177.50    |      173.93     |
>
> By comparing the performance between models tuned from selected subsets and the whole dataset, we observe that less instruction data for better performance still work for various MLLMs. Our findings indicate promising results, suggesting that our method is not only effective with MiniGPT-4 but also exhibits potential for broader applicability.
>
> [1] Zhou, Liu, et al. Lima: Less is more for alignment. arXiv preprint arXiv:2305.11206, 2023.
>
> [2] Bai, et al. Qwen-vl: A frontier large vision-language model with versatile abilities. arXiv preprint arXiv:2308.12966, 2023.
>
> [3] Liu, Li, et al. Visual instruction tuning. arXiv preprint arXiv:2304.08485, 2023.
>
> Q2: There are many systematic methods in the literature that do something similar to select a subset from a dataset based on the model performance on a validation set. A notable one is Meta-Weight-Net [1] based on meta learning. How does this InstructionGPT-4 compare with the Meta-Weight-Net?
>
> A2: Thanks for your suggestion. It's important to clarify that while both our method and Meta-Weight-Net aim to improve model performance through selective data, **they are based on different learning mechanisms**.
> Meta-Weight-Net, as you mentioned, is grounded in meta-learning. It typically involves generating weights for training samples in a classification task, which are then used to adjust the loss function during the training of a model like ResNet. This approach is particularly suited for classification tasks where the output of a MLP can effectively influence the training process of the model.
>
> In contrast, **our work focuses on subset selection for generative tasks within the multimodal LLM domain**. Our approach is not about adjusting weights during training but rather about selecting a subset of data that improves the efficacy of fine-tuning for generative tasks. This involves developing a multimodal data selector that identifies the most impactful data for model alignment, which is a different challenge compared to what Meta-Weight-Net addresses.
>
> Given the generative nature of our task and the multimodal aspects of our model, applying a meta-learning approach like that of Meta-Weight-Net directly to our context would be challenging. The complexity of generative tasks in multimodal settings requires a different strategy, which is what we have developed and focused on.
>
> We have included citations for Meta-Weight-Net in the revised version along with the above detailed discussion.
>
> Besides, we conduct additional experiments on several data pruning methods including EL2N [1] and prototypicality [2], which were proposed in vision modal. We represent our experiment results below:
>
> | Benchmark | InstructionGPT-4 |  EL2N  | Prototypicality | Random |
> |:---------:|:----------------:|:------:|:---------------:|:------:|
> | MME       |      **648.26**      | 627.27 |     569.46      | 527.26 |
> | VQA       |      **22.30**       |  20.62 |     21.83       |  19.87 |
>
>
> We observe that these previous systematic methods for single modal can't achieve competitve performance for multimodal LLM compared to our data selector. It showcases that our novel data selection method for multimodal datasets is quite necessary. We also add these experiments to the revised version along with the above detailed discussion.

---

> ### Author Response · Authors · 2023-11-18
> **Response to Reviewer 61XF (2/3)**
>
> Q3: The method obtained slightly better scores than the MiniGPT4 for some scores, but it is partially weaker than MiniGPT4. Moreover, in the benchmarks MME and MMBench, MiniGPT4 is a weak baseline. The marginal improvement over MiniGPT4 is not convincing.
>
> A3: Thanks for your question. **Our primary goal in this research is to explore and demonstrate the impact of using a smaller but higher-quality dataset on the performance of multimodal LLMs**. Indeed, our findings show that with tuned by only 6% of the original data, InstructionGPT-4 is able to surpass the performance of MiniGPT-4 in most tasks, which is a notable accomplishment. This outcome challenges the conventional notion that more data invariably leads to better model performance, highlighting instead the value of high-quality, well-selected instruction data.
>
> While the improvements over MiniGPT-4 may appear marginal in some aspects, they are still significant in our research objectives. **Our achievement lies in not only surpassing an established baseline but also firstly proposing a novel method for selecting high-quality multimodal data**. This finding has important implications for the efficiency and sustainability of training large LLMs, suggesting that strategic data selection can reduce resource requirements without compromising performance.
>
> Furthermore, we choose MiniGPT-4 for experiments because it is a widely recognized and accessible multimodal LLM, making it a practical choice for benchmarking. Additionally, **we extend our experiments to include Qwen_VL, a state-of-the-art open-sourced model with a strong baseline, to validate our approach across different multimodal LLMs, which is also mentioned in A1 above**.
>
> Q4: If the data selector $F$ takes a data sample as input (see questions below for this), then in the paper the authors actually assign the same label to every data sample in the same subset $D_i$, which does not make sense because the label is due to the contribution of the whole subset (the model performance after being trained on the subset).
>
> A4: Thanks for your comment. **The decision to assign the same label to every data sample in the same subset $D_i$ is indeed a tradeoff, and I'd like to clarify the rationale behind this approach**.
>
> If we were to fine-tune our pre-trained model on each individual data sample and then evaluate the performance of the fine-tuned model to determine quality labels, the process would require training and evaluating thousands of models. This approach would be prohibitively expensive and time-consuming, making it impractical for our purposes. At the same time, for the second-stage fine-tuning of multimodal LLMs, the core purpose is to align the instructions, responses and images. If the model is only fine-tuned on a very small dataset, the model training is not sufficient to obtain good alignment results, and the evaluation will be meaningless at this situation.
>
> Thus, we need to divide the original dataset into several groups. The size of each group cannot be too large or too small. On the one hand, if each group contains too much data, then there will be too few groups to fit an appropriate mapping relationship when training the data selector. On the other hand, if each group contains too little data, then the MLLM cannot be fully fine-tuned for alignment. So in this tradeoff, we choose to divide the data into 30 groups to ensure that there are enough samples during fitting and that the MLLM can achieve good alignment results when fine-tuned on each group.
>
>
> Q5: What features of the data samples did the authors use for the clustering algorithms? Are the features flattened images or some extracted image features?
>
> A5: Thank you for your question. As outlined in Section 2.1 of our paper, **we use image embeddings for the clustering process**. Specifically, we do not use raw, flattened images directly. Instead, we extract and encode image features to create these embeddings, which then serve as the input for our spectral clustering algorithm.
> These image embeddings are derived from a pre-trained ViT, which is capable of converting the visual content of the images into a more abstract and condensed representation. This representation captures the essential characteristics of the images. By using these encoded image embeddings, our spectral clustering algorithm can more effectively categorize the data into ten distinct groups based on the inherent similarities and differences in the visual features.

---

> ### Author Response · Authors · 2023-11-18
> **Response to Reviewer 61XF (3/3)**
>
> Q6: The description in Section 2.3 is confusing. In the "Training" paragraph, it seems that the data selector F maps from the feature of a whole subset to a score, because the index i is for the subsets and it seems to mean yi=F(ei). But in the "Testing" paragraph, it seems that the data selector F maps from the feature of a single data sample to a score. Can the authors clarify it?
>
> A6: Thank you for your inquiry regarding the features used to train the data selector F. Assume that each data point in our dataset is represented by a D-dimensional feature vector. When we form a cluster of K data points, these D-dimensional vectors are concatenated to create a feature vector with a shape of (K, D). **In training the data selector F, we utilize these (K, D) shape vectors**. Each cluster, represented by its (K, D) vector, is associated with a "genius quality label" that reflects the collective quality of all K data points within the cluster. This approach allows us to train the data selector on a rich and detailed representation of data clusters.
>
> Regarding the application of F on a single data point as discussed in Section 2.3, it's important to clarify that F is trained on clusters but is capable of evaluating individual data points as well. **When applied to a single data point, F operates on its (1, D) feature vector**. This flexibility is a key aspect of F, allowing it to function effectively both at the cluster level during training and at the individual data point level during testing.
>
> To address the potential confusion this might cause, we have revised the relevant sections of our paper to more clearly articulate how F is trained on clusters but can also evaluate individual data points. This explanation should make it clearer how F operates across different levels of data granularity.
>
>
>
> Q7: Why did the authors not simply use the quality scores to select a subset for finetuning the final MLLM? It seems that the training data and the testing data for the data selector are the same. What is the meaning of performing the testing if we already have the labels on the data?
>
> A7: Thank you for your question. We do not use the "genuine quality label" as the criterion for data selection because **different data points in the same subset divided by K-Means++ share the same "genuine quality label"**. We need to distinguish these data points in same subsets by conducting a testing stage.
>
> By employing spectral clustering after the data selector has been trained, it helps to prevent the homogenization of selected data and ensures that our model can handle a variety of data patterns. This diversity is crucial for capturing a wide range of data distribution patterns in the multimodal dataset, which can not be adequately represented if we solely rely on "genuine quality labels" for data selection.
>
> We have made efforts to clarify this distinction and the importance of each clustering step in our revised paper.
>
> Q8: What is the relationship between the validation data used to generate the quality labels and the evaluation data? Why could the model performance on validation data be used as an indicator for model performance on the evaluation (test) data?
>
> A8: Thank you for your question. In our study, **the validation data consists of four widely recognized VQA datasets**. We choose these four VQA datasets to produce "genuine quality labels" because they are sufficiently diverse and contain various question-answer pairs.
>
> **The evaluation data, on the other hand, include additional VQA datasets and multiple-choice questions specifically designed for multimodal large language models**. A key dataset we use for evaluation is MME, a standard benchmark in the field. Unlike the validation datasets, MME is not split into training and test sets; it's used entirely for evaluation purposes.
>
> The relationship between the validation and evaluation datasets is founded on the principle of **generalization**. In real-world applications of MLLMs, the specific downstream tasks may not always be known in advance. Therefore, it's crucial to train and validate models on a range of datasets that collectively encompass a broad scope of possible scenarios and tasks. By ensuring our model performs well on the diverse validation datasets, we can reasonably infer that it will also perform effectively on the evaluation datasets, including MME.
>
> Although dividing MME into a training and test set might potentially yield better results due to more targeted training, our approach aims to make multimodal LLM generalized to different tasks. In fact, our validation strategy is designed to strike a balance between specificity and generalizability, ensuring that our model is well-equipped to handle a variety of multimodal tasks.

---

> ### Author Response · Authors · 2023-11-20
> **We would be grateful if you could take a look at the response**
>
> Dear Reviewer 61XF:
>
> We sincerely appreciate your valuable time devoted to reviewing our manuscript. We would like to gently remind you of the **approaching deadline for the discussion phase**. We have diligently addressed the issues you raised in your feedback, providing detailed explinations. For instance, we have straightforwardly applied our data selector for different multimodal LLMs and datasets, which achieve promising performance that demonstrate the generalizability of our data selector. Moreover, we have conducted experiments on several existing data pruning methods that indicate the superiority our multimodal data selection approach. Besides, we clarify that assigning the same label to every data sample in the same subset is reasonable because of a tradeoff. Would you kindly take a moment to look at it?
>
> We are very enthusiastic about engaging in more in-depth discussions with you.

---

> ### Author Response · Authors · 2023-11-22
> **Seeking Your Input on Revised Paper's Alignment with ICLR Standards**
>
> Dear Reviewer 61XF,
>
> As the discussion period approaches its conclusion, **we want to ensure that we have thoroughly addressed all your concerns and that our revised paper fully meets the standards of ICLR**. We would highly value any additional feedback you may provide.
>
> Thank you sincerely for your time and consideration.
>
> Best regards,
>
> The Authors

---

### Official Review · Reviewer_Wn4g · 2023-10-29

**Soundness:** 3 good
**Presentation:** 4 excellent
**Contribution:** 3 good
**Rating:** 6
**Confidence:** 4

**Summary:**

The authors propose a method for automatically selecting high-quality diverse subsets of large multimodal instruction datasets. The method works by using a small network to learn the relationship between a set of indicator features and dataset quality (measured through loss on a validation set). Additionally, clustering is performed before data selection in order to ensure the selected dataset is diverse. The authors evaluate over a number of benchmarks and find that their model trained on only 200 examples outperforms a model trained on all available data (MiniGPT-4) and a model trained on 200 randomly chosen examples, showing the efficacy of their data selection method.

**Strengths:**

- The proposed data selection method is reasonable and the evaluation seems to indicate it is effective. The method itself is reasonably novel, not just adapting a text-only method for data selection to the multimodal space, but also improving on approaches such as InstructMining through additional steps.
- Having a more powerful model trained on less data, and a method for automatically selecting quality data, is interesting and useful for improving overall model performance and reducing the compute required to train quality models.
- The evaluation is reasonably extensive, covering both benchmarks and open-ended (GPT-4-based) evaluation.
- The authors ablate each aspect of their method (clustering, indicators, data selector) and justify each decision made. In particular, the ablations exploring the use of single indicators may be useful for future examining data quality.

**Weaknesses:**

- As far as I can tell, results with the selected data are from single runs. I worry that when selecting a small number of examples, the variance of the results might be quite high. Seeing results over multiple random seeds with regards to the data selection (training and selection itself) would be useful. However, considering the variance over the random selection is not massive, I think it's clear the selection method is effective in most cases.
- The authors only explore one model/architecture, so it is unclear how well their method would apply to larger/different models. In particular, as models scale, they benefit less from instruction data, so it is unclear if the rules for selecting quality data derived from smaller models will transfer to larger models.
- The GPT-4 evaluation is performed over only 60 questions, which seems quite small. In particular, I worry that 60 questions is not enough to thoroughly test the model over a diverse set of queries that may expose gaps in performance due to the small number of examples seen in finetuning compared to a fully-finetuned model. This can be seen e.g. in Table 2, where InstructionGPT-4 performs significantly worse than MiniGPT-4 in commonsense reasoning and OCR tasks.

Overall, I think the work is solid, and presents an interesting and useful method for data selection. It would be nice to see better validation of the method through repeated trials and over different architectures/datasets, but I think the results presented here are solid, with extensive evaluation.

**Questions:**

- Why do you think there is a big drop in MME performance when moving from two layers to three layers in table 5?
- In Figure 3, it appears that MiniGPT outperforms IntructionGPT-4 in a smaller number of tasks, especially looking at the natural relation and object localisation task. Why do you think this is?
- It would be useful and interesting to see when using the data selection method becomes no better than random selection (if this happens) when increasing the number of data points selected (i.e. extending Figure 7 and adding a line for random data selection).

---

> ### Author Response · Authors · 2023-11-18
> **Response to Reviewer Wn4g (1/2)**
>
> Q1: Seeing results over multiple random seeds with regards to the data selection (training and selection itself) would be useful.
>
> A1: Thanks for your suggestion. Our main motivation is to prove that less but high-quality instruction data can outperform the whole dataset. In particular, we want to find out **if there exists a subset making InstructionGPT-4 achieve better performance**. As a result, our proposed method indeed shows the existence of the subset.
>
> Moreover, in response to your suggestion, we conduct additional experiments to try different random seeds. These experiments help us evaluate the stability and generalizability of our selection method under different initialization conditions. Our experimental results are shown below:
>
> | Benchmark |  Seed1  |  Seed2  |  Seed3  | MiniGPT-4 |
> |:---------:|:-------:|:-------:|:-------:|:---------:|
> | MME       |  648.26 |  637.58 |  628.99 |   625.20  |
> | VQA       |  22.30  |  21.78  |  22.14  |   20.54   |
>
>
> We observe that the performance variance across different seeds is not obvious. This consistency of the performance underscores the effectiveness of our data selection method, suggesting that it is not overly sensitive to the initial random seed and can reliably identify high-quality data subsets across different scenarios. We also add these experiments to the revised version.
>
> Q2: The authors only explore one model/architecture, so it is unclear how well their method would apply to larger/different models. In particular, as models scale, they benefit less from instruction data, so it is unclear if the rules for selecting quality data derived from smaller models will transfer to larger models.
>
> A2: Thank you for your comments. Following LIMA [1] that proposes "less is more for alignment" only using LLaMA in textual modal, our main motivation is to prove that less but high-quality instruction data can outperform the whole dataset for multimodal LLMs, which is also mentioned in A1 above. Besides, we want to focus on MiniGPT-4 because it's the first and most popular multimodal model utilizing LLM.
> To address your concern about scalability and generalizability, we also extend our investigation to include additional models and datasets. **Notably, we have applied our method to Qwen_VL [2] and the detail_10k [3] dataset without retraining the data selector**.
> Here are our experimental results:
>
> | detail_10k   | Qwen_VL (1k) | Qwen_VL (2k) | Qwen_VL (10k) | MiniGPT-4 (1k) | MiniGPT-4 (2k) | MiniGPT-4 (10k) |
> |:------------:|:------------:|:------------:|:-------------:|:--------------:|:--------------:|:---------------:|
> | MME          |    1802.50   |   **1806.81**|    1769.28    |   **614.04**   |      608.54    |      604.47     |
> | Perception   |    1423.57   |   **1426.10**|    1398.92    |   **434.04**   |      431.04    |      428.54     |
> | Cognition    |    378.93    |   **380.71** |    370.36     |   **180.00**    |      177.50    |      173.93     |
>
> By comparing the performance between models tuned from selected subsets and the whole dataset, we observe that less instruction data for better performance still work for various MLLMs. Our findings indicate promising results, suggesting that our method is not only effective with MiniGPT-4 but also exhibits potential for broader applicability.
>
> [1] Zhou, Liu, et al. Lima: Less is more for alignment. arXiv preprint arXiv:2305.11206, 2023.
>
> [2] Bai, et al. Qwen-vl: A frontier large vision-language model with versatile abilities. arXiv preprint arXiv:2308.12966, 2023.
>
> [3] Liu, Li, et al. Visual instruction tuning. arXiv preprint arXiv:2304.08485, 2023.
>
> Q3: The GPT-4 evaluation is performed over only 60 questions, which seems quite small. In particular, I worry that 60 questions is not enough to thoroughly test the model over a diverse set of queries that may expose gaps in performance due to the small number of examples seen in finetuning compared to a fully-finetuned model.
>
> A3: Thank you for your comments. Firstly, in our evaluation, we want to cover both close-ended and open-ended benchmarks. For open-ended benchmarks, **LLaVA-Bench is the only one requiring GPT-4 for evaluation proposed during our work to our best knowledge**. Secondly, while LLaVA-Bench only contains 60 questions, it includes suffcient senarios such as indoor and outdoor scenes, memes, paintings, and sketches, and each image is associated with a carefully designed selection of high-quality questions. Thirdly, LLaVA-Bench is widely adopted by most recent related works, such as [1] and [2]. In addition, besides LLaVA-Bench, we also conduct sufficient closed-ended experiments, showing the effectiveness of our proposed method. We will try other multimodal open-ended benchmarks as future work.
>
> [1] Sun, Shen, et al. Aligning large multimodal models with factually augmented rlhf. arXiv preprint arXiv:2309.14525, 2023.
>
> [2] Liu, Li, et al. Improved baselines with visual instruction tuning. arXiv preprint arXiv:2310.03744, 2023.

---

> ### Author Response · Authors · 2023-11-18
> **Response to Reviewer Wn4g (2/2)**
>
> Q4: Why do you think there is a big drop in MME performance when moving from two layers to three layers in table 5?
>
> A4: Thanks for your question. Given a fixed size of training set, **more layers containing more parameters tend to overfit**. Therefore, there will be a drop in MME performance when the number of layers increases. In our specific case, we cluster the datasets into 30 groups for training so that the training set is extremely small, and thus a drop in MME performance was observed from two layers to three layers.
>
> Q5: In Figure 3, it appears that MiniGPT outperforms IntructionGPT-4 in a smaller number of tasks, especially looking at the natural relation and object localisation task. Why do you think this is?
>
> A5: Indeed, in a limited set of tasks, such as natural relation and object localization, MiniGPT-4 appears to outperform InstructionGPT-4. This outcome aligns with our core objective, which is to demonstrate that **InstructionGPT-4 can surprisingly achieve superior performance in most tasks using a significantly reduced dataset** (only 6% of the total data).
>
> When we increase the size of selected data, InstructionGPT-4 does start to lead in these specific tasks. However, our primary goal is not to make InstructionGPT-4 outperform MiniGPT-4 across every individual task, but rather to showcase the effectiveness of using a well-curated subset for training large multimodal models. This is a significant finding in itself, as it suggests that careful data selection can lead to substantial improvements in model performance without the need for extensive datasets.
>
> Additionally, it is important to recognize that large multimodal models may not excel in all types of tasks. For instance, in tasks like OCR, we observe that when these models integrate image information with large language model capabilities, there can be a loss of detail due to the compression of image tokens. This results in a reduced granularity of information, which in turn weakens their OCR performance. This specific limitation highlights a current challenge in the field and suggests a direction for future research.
>
> Q6: It would be useful and interesting to see when using the data selection method becomes no better than random selection (if this happens) when increasing the number of data points selected (i.e. extending Figure 7 and adding a line for random data selection).
>
> A6: Thank you for this suggestion. We have incorporated this experiment into our revised paper, extending the analysis in Figure 7 to include a comparison with random data selection. We also display the results below:
>
> | Data Size             |    50   |   100  |   150  |   200  |   250  |   300  |   400  |   600  |   800  |  1000  |
> |:---------------------:|:-------:|:------:|:------:|:------:|:------:|:------:|:------:|:------:|:------:|:------:|
> | Data Selector (MME)   |  515.98 | 599.59 | 617.17 | 648.26 | 648.97 | 649.68 | 651.83 | 643.00 | 622.21 | 598.44 |
> | Random Selection (MME)|  435.94 | 535.93 | 523.63 | 527.26 | 602.12 | 615.82 | 608.07 | 612.10 | 622.19 | 616.75 |
> | Data Selector (VQA)   |  22.75  |  22.03 |  21.99 |  22.30 |  22.17 |  21.88 |  21.92 |  21.67 |  21.00 |  20.70 |
> | Random Selection (VQA)|  20.30  |  20.62 |  19.83 |  19.87 |  20.21 |  20.23 |  21.36 |  20.73 |  20.38 |  20.68 |
>
>
> Our findings reveal that our data selection method outperforms random selection in all cases except the size of 1000.
> Through our observation, there is an interesting trend as the size of the selected subset increases: the performance gap between our method and random selection begins to narrow.
>
> This decrease in the performance differential can be attributed to the nature of our data selection process. As the size of the subset selected by our method increases, it inevitably starts to include more data points of lower quality. This inclusion of lower-quality data diminishes the overall effectiveness of the selected subset, thereby reducing the gap in performance compared to a randomly selected subset.
>
> Despite this trend, it is important to note that our data selection method continually maintains a performance advantage over random selection in most cases. This finding underscores the efficacy of our approach, particularly when working with smaller subsets. It also highlights a key insight: the quality of data, not just the quantity, is crucial for improving model performance.
>
> We believe these results add a valuable dimension to our analysis, providing a clearer understanding of the scalability and efficiency of our data selection method. The revised paper includes a detailed discussion of these findings and their implications for the use of data selection in training large multimodal models.

---

> ### Author Response · Authors · 2023-11-20
> **We would be grateful if you could take a look at the response**
>
> Dear Reviewer Wn4g:
>
> We sincerely appreciate your valuable time devoted to reviewing our manuscript. We would like to gently remind you of the **approaching deadline for the discussion phase**. We have diligently addressed the issues you raised in your feedback, providing detailed explanations. For instance, we have elucidated that our approach is stable when using multiple random seeds with regards to the data selection. Moreover, we have conducted experiments on several existing data pruning methods that indicate the superiority our multimodal data selection approach. Besides, we conduct experiments on increasing the number of data points selected with detailed analysis. Would you kindly take a moment to look at it?
>
> We are very enthusiastic about engaging in more in-depth discussions with you.

---

> > ### Comment · Reviewer_Wn4g · 2023-11-20
> > **Re: Response**
> >
> > Hi, thank you for your detailed response, and the number of detailed experiments.
> >
> > Seed results: I'm not convinced there isn't much variance here. The range of MME performance across seeds is quite large (stdev of 9.7). The variance with VQA seems better, which makes me wonder if the data selection method is somehow biased towards VQA tasks (or maybe the total size of the VQA eval set is just much bigger?).
> >
> > detail_10k results - Thank you for these, the results here seem good, although I would like to similarly see multiple runs and maybe more baselines.
> >
> > Why does MiniGPT outperform InstructionGPT in some tasks - I get that using less data for good overall results is interesting and useful, but I think it's important and useful to discuss where and why your approach falls behind. Perhaps your selection mechanism is biased towards certain tasks, or maybe something about those tasks clashes with the data needs of other tasks that is hard to resolve with a small data budget.
> >
> > Increasing the data points selected - Thank you for this result, this is interesting and I think helps suggest that your data selection method is performing well compared to random selection. It would be good to see something similar to this for other baselines.
> >
> > I've read the other reviews and responses carefully, and am keeping my score. I think that work in data selection is generally interesting and useful, and the results here seem reasonable compared to data selection. I agree with other reviewers that more thorough comparisons with baselines and more analysis of what the data selector is doing would improve the work.

---

> ### Author Response · Authors · 2023-11-22
> **Further Response to Reviewer Wn4g**
>
> Q1: Seed results: I'm not convinced there isn't much variance here. The range of MME performance across seeds is quite large (stdev of 9.7). The variance with VQA seems better, which makes me wonder if the data selection method is somehow biased towards VQA tasks (or maybe the total size of the VQA eval set is just much bigger?).
>
> A1: Thanks for your question. Our main motivation is to prove that less but high-quality instruction data can outperform the whole dataset. In particular, we want to find out if there exists a subset making InstructionGPT-4 achieve better performance. As a result, our proposed method indeed shows the existence of the subset.
>
> In addition, we add one more seed and found that the stdev is 8.3 and the average is 641.02, which means that there is over 90% probability that InstructionGPT-4 outperforms MiniGPT-4 from the empirical rule.
>
> Moreover, the size of MME is relatively small compared to other VQA datasets. So the variance is bigger than testing on VQA.
>
> | Benchmark | Seed1 | Seed2 | Seed3 | Seed4 | Average | MiniGPT-4 |
> | --- | --- | --- | --- | --- | --- | --- |
> | MME | 648.26 | 637.58 | 628.99 | 649.25 | 641.02 (8.32) | 625.20 |
> | VQA | 22.30 | 21.78 | 22.14 | 21.98 | 22.05 (0.19) | 20.54 |
>
>
>
> Q2: detail_10k results - Thank you for these, the results here seem good, although I would like to similarly see multiple runs and maybe more baselines.
>
> A2: Thanks for your suggestion. We are currently conducting these new experiments. We will add multiple runs and more baselines as soon as we complete.
>
> Q3: Why does MiniGPT outperform InstructionGPT in some tasks - I get that using less data for good overall results is interesting and useful, but I think it's important and useful to discuss where and why your approach falls behind. Perhaps your selection mechanism is biased towards certain tasks, or maybe something about those tasks clashes with the data needs of other tasks that is hard to resolve with a small data budget.
>
> A3: Thanks for your comment. The analysis for MLLM performance towards certain tasks is a current challenge. For instance, in tasks like OCR, researchers observe that increasing training data is necessary because when MLLMs integrate image information with large language model capabilities, which can result in a loss of detail due to the compression of image tokens [1]. Your suggestion for detailed analysis of certain tasks can be a direction for future research.
>
> [1] Han, Zhang, et al. Imagebind-llm: Multi-modality instruction tuning. arXiv preprint arXiv:2309.03905, 2023.
>
> Q4: Increasing the data points selected - Thank you for this result, this is interesting and I think helps suggest that your data selection method is performing well compared to random selection. It would be good to see something similar to this for other baselines.
>
> A4: Thanks for your suggestion. We will add other baselines as soon as we complete these experiments.

---

> ### Author Response · Authors · 2023-11-22
> **Seeking Your Input on Revised Paper's Alignment with ICLR Standards**
>
> Dear Reviewer Wn4g,
>
> As the discussion period approaches its conclusion, **we want to ensure that we have thoroughly addressed all your concerns and that our revised paper fully meets the standards of ICLR**. We would highly value any additional feedback you may provide.
>
> Thank you sincerely for your time and consideration.
>
> Best regards,
>
> The Authors

---

### Official Review · Reviewer_dCjW · 2023-10-31

**Soundness:** 2 fair
**Presentation:** 2 fair
**Contribution:** 2 fair
**Rating:** 1
**Confidence:** 5

**Summary:**

The authors performed a series of steps aimed to selecting training data for multimodal large langauge models (MLLMs). They first extracted freatures using a number of heuristics, such as CLIP score, length of the answer, and so on. After that, they cluster the data, train miniGPT-4 on each cluster, and test the results on VQA datasets to get "genuine quality labels". In the next step, they train a network to predict quality labels from the feature vectors. Finally, they recluster the data, and use the trained network to select the best data points.

The authors compared against the original miniGPT and a random baseline for data selection. The selected 200 data points outperform both.

**Strengths:**

The paper achieves good results. 200 data points they selected did lead to performance improvements over the original miniGPT-4.

The paper contains few grammatical errors. I thank the authors for proofreading the paper.

**Weaknesses:**

The most important weakness of the paper is the lack of baselines. Many data quality and data pruning metrics have been proposed over the years. The authors should consider comparing against Influence Functions, Data Shapley, HyDRA, Diverse Ensembles [1], prototypicality [2], etc. A random baseline is way too weak.

1. Kristof Meding, Luca M. Schulze Buschoff, Robert Geirhos, and Felix A. Wichmann. Trivial or impossible—dichotomous data difficulty masks model differences (on ImageNet and beyond). In International Conference on Learning Representations, 2022.

2. Sorscher, Ben, et al. Beyond neural scaling laws: beating power law scaling via data pruning. NeurIPS. 2022.

It is not clear how the propose method could scale or generalize to different datasets and models, as it is only tested on a single dataset and a single MLLM. It is also not clear why only four VQA datasets are selected to produce "genuine quality labels", whereas the model is evaluated on bigger benchmarks.

The writing contains a number of vague descriptions and logic disconnects (see below and the questions). The authors seem to be in a habit of fancy word choices, which may be the result of some large language model. However, word choices, however literary, cannot compensate for logic issues.

- Page 3: ... steer multimodal language models in learning to generate responses in a particular manner --> what manner is that?
- Page 4: .. can more completely cover the various aspects of multimodal data quality -> More than what? What exactly are the aspects of multimodal data quality?
- Page 4: In equation (1), the dimensionality reduction (DR) technique is written as P(f(x_image), g(x_response)). Does it mean that DR is performed on the concatenation of image features and text features?
- Page 5: I don't understand what is x in the unnumbered equations of Section 2.3. Is x a data point or a cluster? Btw, it's a good idea to number all equations, in case your reviewers want to refer to them.
- Page 5: What features is the data selector F trained with? The text says "we concatenate these embeddings into a single composite embedding". If a data point has D-dimensional features, and a cluster contains K data points, do we have a KD-dimensional feature vector? If that's the case, how can we apply F on a single data point, as implied in the later part of Section 2.3?

**Questions:**

- How do you make sure K-means and spectral clustering return clusters of equal size?
- What is the purpose of reclustering the data points? Why not use the "genuine quality label" as the criterion for data selection?
- Why is length such an important feature? Although the feature is very simple, it alone achieves the third best result in the ablation study of Table 5, only inferior to CLIP and the whole feature set. Does this imply the data selection process is utilizing a shortcut feature overfitted to the test?

---

> ### Author Response · Authors · 2023-11-18
> **Response to Reviewer dCjW (1/4)**
>
> Q1: The most important weakness of the paper is the lack of baselines. Many data quality and data pruning metrics have been proposed over the years. The authors should consider comparing against Influence Functions, Data Shapley, HyDRA, Diverse Ensembles [1], prototypicality [2], etc. A random baseline is way too weak.
>
> A1: Thanks for your suggestion. We would like to highlight that **we are the first to propose a data selection method working on multimodal LLM for generative tasks**. Since there are no other data seletion methods for multimodal LLMs, we have to compare our method with single-modal baselines in our ablation study. In particular, **we compare our proposed selector with the recent methods such as Alpagasus (GPT Score only) in Table 5 and Instruction Mining (Linear Selector) in Figure 5**, demonstating that data selected by our multimodal selector works much better than these two methods.
> While our data selection method and data pruning methods both aim to reduce the size of training data, we would like to emphasize several important differences.
> Our data selection method is used for multimodal LLMs in generative tasks. In contrast, data pruning methods are mostly used for vision models (e.g., ResNet, ViT) in classification tasks.
> Moreover, in response to your suggestion, we conduct additional experiments on data pruning methods including EL2N [1] and prototypicality [2]. We represent our experiment results below:
>
> | Benchmark | InstructionGPT-4 |  EL2N  | Prototypicality | Random |
> |:---------:|:----------------:|:------:|:---------------:|:------:|
> | MME       |      **648.26**      | 627.27 |     569.46      | 527.26 |
> | VQA       |      **22.30**       |  20.62 |     21.83       |  19.87 |
>
>
>
> We observe that these previous methods can't achieve competitve performance for multimodal LLM compared to our data selector. It showcases that our novel data selection method for multimodal datasets is quite necessary. We also add these experiments to the revised version along with the above detailed discussion.
>
> [1] Paul, Ganguli, Dziugaite G K. Deep learning on a data diet: Finding important examples early in training. NeurIPS, 2021.
>
> [2] Sorscher, Ben, et al. Beyond neural scaling laws: beating power law scaling via data pruning. NeurIPS, 2022.
>
> Q2: It is not clear how the propose method could scale or generalize to different datasets and models, as it is only tested on a single dataset and a single MLLM.
>
> A2: Thank you for your comments. Following LIMA [1] that proposes "less is more for alignment" only using LLaMA in textual modal, our main motivation is to prove that less but high-quality instruction data can outperform the whole dataset for multimodal LLMs. In particular, we want to **find out if there exists a subset making InstructionGPT-4 achieve better performance**. As a result, our proposed method indeed shows the existence of the subset. Besides, we want to focus on MiniGPT-4 because it's the first and most popular multimodal model utilizing LLM.
> To address your concern about scalability and generalizability, we also extend our investigation to include additional models and datasets. **Notably, we have applied our method to Qwen_VL [2] and the detail_10k dataset [3] without retraining the data selector**.
> Here are our experimental results:
>
> | detail_10k   | Qwen_VL (1k) | Qwen_VL (2k) | Qwen_VL (10k) | MiniGPT-4 (1k) | MiniGPT-4 (2k) | MiniGPT-4 (10k) |
> |:------------:|:------------:|:------------:|:-------------:|:--------------:|:--------------:|:---------------:|
> | MME          |    1802.50   |   **1806.81**|    1769.28    |   **614.04**   |      608.54    |      604.47     |
> | Perception   |    1423.57   |   **1426.10**|    1398.92    |   **434.04**   |      431.04    |      428.54     |
> | Cognition    |    378.93    |   **380.71** |    370.36     |   **180.00**    |      177.50    |      173.93     |
>
>
> By comparing the performance between models tuned from selected subsets and the whole dataset, we observe that less instruction data for better performance still work for various MLLMs. Our findings indicate promising results, suggesting that our method is not only effective with MiniGPT-4 but also exhibits potential for broader applicability.
>
> [1] Zhou, Liu, et al. Lima: Less is more for alignment. arXiv preprint arXiv:2305.11206, 2023.
>
> [2] Bai, et al. Qwen-vl: A frontier large vision-language model with versatile abilities. arXiv preprint arXiv:2308.12966, 2023.
>
> [3] Liu, Li, et al. Visual instruction tuning. arXiv preprint arXiv:2304.08485, 2023.

---

> ### Author Response · Authors · 2023-11-18
> **Response to Reviewer dCjW (2/4)**
>
> Q3: It is also not clear why only four VQA datasets are selected to produce "genuine quality labels", whereas the model is evaluated on bigger benchmarks.
>
> A3: Thanks for your question. We choose these four VQA datasets to produce "genuine quality labels" because **they are sufficiently diverse and contain various question-answer pairs**. In particular, GQA focuses on reasoning skills and combined language understanding skills; IconQA requires perceptual skills such as object recognition and text understanding; OKVQA is a large-scale dataset requiring external knowledge; ScienceQA can well diagnose whether the multimodal LLM has multi-step reasoning capabilities and interpretability. These four datasets are wide enough to cover multiple aspects of multimodal tasks. Thus, we choose to select these four datasets to produce "genuine quality labels".
>
> Q4: Page 3: ... steer multimodal language models in learning to generate responses in a particular manner --> what manner is that?
>
> A4: Thanks for your question. In our paper, **the "manner" refers to the stylistic and structural characteristics of the generated content**, which are significantly influenced by the the instruction data. Instruction data can greatly affect models' generation during instruction tuning. Thus, **the manner of the generated content of the multimodal model is closely related to the manner of the instruction data**. To illustrate, if the instruction data mostly consist of lengthy and complex sentences, the model is more possible to generate responses that follow this style -- extensive and intricate. Conversely, if the instruction data are composed of concise, straightforward sentences, the model tends to produce briefer and more direct responses.  We have edited this to a clearer description in the revised version.
>
> Q5: Page 4: .. can more completely cover the various aspects of multimodal data quality -> More than what? What exactly are the aspects of multimodal data quality?
>
> A5: Thanks for your question. In our paper,  it means that **more than using only one quality method to define multimodal data quality**, which is mentioned behind this sentence: "Using a single score to filter data can be useful, but it may not provide a comprehensive measure of data quality. Therefore, it is necessary to combine multiple indicators as an embedding to assess data quality collectively". When we state that our method "more completely covers the various aspects of multimodal data quality", we are comparing it to approaches that rely on a single metric for data quality assessment. Specifically, some existing methods, like Alpagasus [1], use the GPT score to evaluate textual data quality. However, we argue that multimodal data quality is multi-faceted and cannot be accurately captured by a single metric. To address this problem, our approach incorporates a combination of metrics: CLIP score, length score, reward score, and GPT score. This multi-metric method allows for a more complete evaluation of multimodal data quality.
>
> [1] Chen, Li, et al. Alpagasus: Training a better alpaca with fewer data. arXiv preprint arXiv:2307.08701, 2023.
>
> Q6: Page 4: In equation (1), the dimensionality reduction (DR) technique is written as P(f(x_image), g(x_response)). Does it mean that DR is performed on the concatenation of image features and text features?
>
> A6: Thanks for your question. To clarify, in our method, **dimensionality reduction is applied separately to the image and text features before concatenation**. Once these features have been independently reduced to the same dimension, we then concatenate them. This process is visually represented in Figure 2 of our paper. This approach ensures that both image and text features are compressed and represented before they are combined, which helps to preserve the unique characteristics of both visual and textual data and allow for an effective integration of multimodal data.
>
> We have amended the relevant section to better explain the sequential steps of dimensionality reduction followed by concatenation to clear up any ambiguities regarding our process.
>
> Q7: Page 5: I don't understand what is x in the unnumbered equations of Section 2.3. Is x a data point or a cluster?
>
> A7: Thanks for your question. From our paper in Section 2.3 Testing: "Given a multimodal dataset D of triplets x = (image, instruction, answer)", **x refers to an individual data point in our multimodal dataset D**. Specifically, each data point x is a triplet containing an image, an instruction, and an answer.

---

> ### Author Response · Authors · 2023-11-18
> **Response to Reviewer dCjW (3/4)**
>
> Q8: Page 5: What features is the data selector F trained with? The text says "we concatenate these embeddings into a single composite embedding". If a data point has D-dimensional features, and a cluster contains K data points, do we have a KD-dimensional feature vector? If that's the case, how can we apply F on a single data point, as implied in the later part of Section 2.3?
>
> A8: Thank you for your inquiry regarding the features used to train the data selector F. Each data point in our dataset is represented by a D-dimensional feature vector. When we form a cluster of K data points, these D-dimensional vectors are concatenated to create a feature vector with a shape of (K, D). **In training the data selector F, we utilize these (K, D) shape vectors**. Each cluster, represented by its (K, D) vector, is associated with a "genius quality label" that reflects the collective quality of all K data points within the cluster. This approach allows us to train the data selector on a rich and detailed representation of data clusters.
> Regarding the application of F on a single data point as discussed in Section 2.3, it's important to clarify that F is trained on clusters but is capable of evaluating individual data points as well. **When applied to a single data point, F operates on its (1, D) feature vector**. This flexibility is a key aspect of F, allowing it to function effectively both at the cluster level during training and at the individual data point level during testing.
>
> To address the potential confusion this might cause, we have revised the relevant sections of our paper to more clearly articulate how F is trained on clusters but can also evaluate individual data points. This explanation should make it clearer how F operates across different levels of data granularity.
>
> Q9: The writing contains a number of vague descriptions and logic disconnects (see below and the questions). The authors seem to be in a habit of fancy word choices, which may be the result of some large language model. However, word choices, however literary, cannot compensate for logic issues.
>
> A9: Thanks for your comment. We have undertaken a thorough review of our manuscript, focusing specifically on areas where the descriptions were vague or the logic was unclear. We also pay particular attention to the sections and points you've highlighted, ensuring that our arguments are presented clearly and logically.  We believe these revisions will address the issues you've raised and enhance the overall readability and coherence of our paper.
>
> Q10: How do you make sure K-means and spectral clustering return clusters of equal size?
>
> A10:  Thanks for your question. In the selector's training stage, we observe that the data quantity in each group from K-means clustering is roughly the same, while other clustering algorithms results in more significant differences in group sizes. This is a characteristic of K-means, as it partitions data into clusters that minimize the variance within each cluster. To acquire genuine quality labels (which necessitate training a multimodal model on each cluster group for evaluation) without letting the quantity of data impact the capabilities of the trained multimodal models, we employ a simple post-processing technique. This involves identifying clusters with either excessively high or low sample counts, and **redistributing some samples from the larger clusters to the smaller ones based on their distance to different cluster centroids**, thereby equalizing the number of samples in each cluster.
>
> In the testing stage, we use spectral clustering to select a diverse set of data points. Unlike in the training stage, the requirement here is not to have clusters of equal size but rather to select the top 200 data points that best represent the diversity of the dataset. Hence, in this stage, the equal size of clusters is not a concern as our focus shifts to ensuring the diversity and representativeness of the selected data.
>
> We have revised our paper to clearly explain this differentiation in the use of clustering techniques between the training and testing stages, and how we ensure appropriate cluster sizes and data diversity in each stage.

---

> ### Author Response · Authors · 2023-11-18
> **Response to Reviewer dCjW (4/4)**
>
> Q11: What is the purpose of reclustering the data points? Why not use the "genuine quality label" as the criterion for data selection?
>
> A11: Thank you for your question, which we address in Appendix A.3 of our paper. The primary purpose of reclustering data points using **spectral clustering** (denoted as Λ) in the data selection stage is to **ensure the diversity of the selected data**. This diversity is crucial for capturing a wide range of data distribution patterns in the multimodal dataset, which can not be adequately represented if we solely rely on "genuine quality labels" for data selection.
>
> To elaborate, the initial clustering using **K-Means++** (denoted as Γ) serves to **divide the data into subsets for obtaining "genuine quality labels"**. These labels are assigned based on the collective characteristics of data points within each subset. While these labels are valuable for assessing overall data quality, they may not fully capture the diversity within the dataset. This is where spectral clustering comes into play.
>
> We do not use the "genuine quality label" as the criterion for data selection because **different data points in the same subset divided by K-Means++ share the same "genuine quality label"**. We need to distinguish these data points in same subsets by conducting a testing stage. By employing spectral clustering after the data selector has been trained, it helps to prevent the homogenization of selected data and ensures that our model can handle a variety of data patterns.
>
> In addition, **we have conducted detailed experiments to study the effectiveness of applying spectral clustering within the data selector mechanism in Figure 5**. These experiments demonstrate the significance of maintaining diverse vision-language instruction data and validate the necessity of the reclustering process.
> We have made efforts to clarify this distinction and the importance of each clustering step in our revised paper, ensuring a clearer understanding of our methodology and the rationale behind the use of different clustering techniques.
>
> Q12: Why is length such an important feature? Although the feature is very simple, it alone achieves the third best result in the ablation study of Table 5, only inferior to CLIP and the whole feature set. Does this imply the data selection process is utilizing a shortcut feature overfitted to the test?
>
> A12: Thank you for your question. The significance of the length feature in our study indicates that there is a meaningful correlation between the length of the data and its quality or utility for our specific task.
>
> However, **strong correlation and shortcuts are two different things**. Our experimental results in Table 5 only demonstrate strong correlation between the length indicator and data quality, but they do not confirm whether the combination of the four features results in learning shortcuts. Solely based on this experiment, it's inconclusive. We cannot assert that the learning of our data selector is focused on length just because the performance of using all the features together is similar to using only length on a single dataset.
>
> In fact, to properly clarify that the effectiveness of the length feature doesn't imply an overfitting issue, we should refer to experiments in Tables 2, 3, and 4, which show the evaluation across multiple datasets with significant domain gaps. To ensure robustness against overfitting, we've employed a diverse set of datasets for training and testing. Specifically, we use four different VQA datasets for training. For testing, we apply our method to other VQAs, MME, and MMBench datasets. The significant domain gap between these datasets helps mitigate the risk of overfitting to a particular data characteristic. We observe that InstructionGPT-4 outperforms MiniGPT-4 in all these tasks. It can be concluded that InstructionGPT-4 still demonstrates a strong generation ability to out-domain evaluation datasets.
>
> In conclusion, **as there is no overfitting observed across these datasets, it indicates that length is not a shortcut**.
> In light of your query, we have added additional explanations in our revised paper to articulate more clearly why the length feature is impactful and to assure readers that our results aren’t a product of overfitting to test data, but a reflection of genuine correlations found in diverse datasets.

---

> ### Author Response · Authors · 2023-11-20
> **We would be grateful if you could take a look at the response**
>
> Dear Reviewer dCjW:
>
> We sincerely appreciate your valuable time devoted to reviewing our manuscript. We would like to gently remind you of the **approaching deadline for the discussion phase**. We have diligently addressed the issues you raised in your feedback, providing detailed explanations. For instance, we have elucidated that our approach is different from the previous data pruning methods because we are the first to focus on multimodal data selection in generative tasks. Moreover, we have conducted experiments on several existing data pruning methods that indicate the superiority our multimodal data selection approach. Besides, by straightforwardly applying our data selector for different multimodal LLMs and datasets, we have successfully achieved promising performance that demonstrate the generalizability of our data selector. Would you kindly take a moment to look at it?
>
> We are very enthusiastic about engaging in more in-depth discussions with you.

---

> ### Comment · Reviewer_dCjW · 2023-11-20
> **Thank you for the new results**
>
> I went through the author response. I thank the authors for running new experiments under a tight timeline and their willingness to engage in discussions. The new results look promising.
>
> However, I remain unconvinced that the proposed approach represents an effective solution to the data pruning problem in vision-language LLMs for the following reasons.
>
> 1. The list of baselines is still rather short and only scratches the surface of the existing literature. The authors presented two simple baselines for data pruning. This is understandable given limited time. However, we are not reviewing for "what can be reasonable done in a week", but if this paper meets the bar of solid scientific contribution. Many widely known baselines, such as influence functions, have not been considered. In addition, little experimental detail has been provided in the updated version, so it is impossible to verify if the experiments performed under time pressure are correct.
>
> 2. The algorithm seems rather ad-hoc and relies on significant manual intervention.
>   * First, the authors chose four datasets manually to produce "genuine quality labels". It is possible that these four datasets are representative of the data in the tests. If the algorithm critically relies on the choice of these four datasets, then we are not sure if the credit should go to the algorithm or to the manual selection of the datasets.
>   * Second, the authors needed manual intervention on the results of the clustering algorithm.
>   * Third, there is little understanding of what the data selection process is doing. There is no theoretical analysis nor any form of intuition. The data selector heavily relies on the length factor. It is possible that the training data are mismatched in length with the quality label datasets, so length becomes an important factor. In that case, it is hard to imagine that this would generalize to other datasets and other models.
>
> It is important for the authors to make unambiguous claim regarding their contribution. If they are claiming high performance, then perhaps it is ok to have little understanding of how their technique works, as long as performance is very high (SOTA performance, for example). If they are claiming a sound algorithm, then we do need to understand why the algorithm is sound. I am not convinced that the authors have achieved either objective.
>
> The same issue happens with the input to F, which has variable dimensionality. The authors have not explained why the network can accept input with different dimensions. It may work as if a convolutional network is used (or self-attention, which is basically temporal convolution) but the text says it could also be an MLP or linear regression in the ablation. Rather than explaining the reason, they verbally guarantee it. This seems to present the reviewers with a binary choice: either we take their word for it or we do not. With all due respect, I cannot accept this type of paper.

---

> > ### Author Response · Authors · 2023-11-22
> > **Further Response to Reviewer dCjW (1/3)**
> >
> > Q1: The list of baselines is still rather short and only scratches the surface of the existing literature. The authors presented two simple baselines for data pruning. This is understandable given limited time. However, we are not reviewing for "what can be reasonable done in a week", but if this paper meets the bar of solid scientific contribution. Many widely known baselines, such as influence functions, have not been considered. In addition, little experimental detail has been provided in the updated version, so it is impossible to verify if the experiments performed under time pressure are correct.
> >
> > A1: First of all, we would like to clarify that **the motivation behind our work is to investigate if the "less is more" phenomenon observed on single-modal LLMs still holds for multi-modal LLMs**. In particular, we want to show if there exists a subset of the instruction dataset (as smaller as better) such that the model trained on this subset can achieve comparable performance on the evaluation tasks. **Since the evaluation tasks have different data distributions and different task types from the training task (e.g., the evaluation task is yes/no question, while the selector training task is general VQA), the smaller training loss does not directly indicate better performance on the evaluation task. However, previous data pruning method mainly rely on the model loss from the training dataset, which would be unfair to them if they were used directly**. Alternatively, a small number of data pruning methods requires access to the evaluation metric, which involve such extensive computations that they are impractical for use with large models. Thus, **we design a learning-based method by formulating a set of indicators for assessment and train a neural network as a data selector to fit the indicators to the genuine quality labels for selection**.
> > Moreover, **we would like to clarify again that previous data pruning methods have several important differences from our proposed method**. While data pruning is a model performance driven method, we adopt a learning-based method by training a data selector, which can be applied to new datasets independent of certain models. Besides, most of data pruning methods focus on optimizing models based on the loss, while we extract the multimodal data indicators to train the selector based on MLLM performance, which can reduce the gap of data distribution in evaluation set. Furthermore, data pruning aims at cutting off bad data for training, while we concentrate on data mining for selecting a subset with least and best data.
> >
> > Thus, it is not fair to compare our method with other traditional data pruning methods. These data pruning methods either focus on reducing data based on model loss on training set that remain gaps in data distributions and task types on evalaution, or demand large computation resource to evaluate models on each step of pruning iteration (e.g., Data Shapley). We also add more details about our experiments on EL2N and prototypicality to verify the correctness in the new revision.
> >
> > Q2: First, the authors chose four datasets manually to produce "genuine quality labels". It is possible that these four datasets are representative of the data in the tests. If the algorithm critically relies on the choice of these four datasets, then we are not sure if the credit should go to the algorithm or to the manual selection of the datasets.
> >
> > A2: To clarify, **the choice of these four datasets and our proposed algorithm are both important for the whole data selection**. The absence of either would be detrimental. If the data of chosen VQA is inadequate, the results will be biased. If the selection method is inadequate, the performance will suffer. Our experimental results demonstrate that across multiple datasets, performance is consistently strong without bias.

---

> > > ### Comment · Reviewer_dCjW · 2023-11-22
> > >
> > > I am not interested in further discussion, as my stance has been made abundantly clear. If I may, I would suggest to the authors that every paper review is fundamentally a value judgment, where people naturally will disagree. The purpose of the rebuttal process is to clear any factual misunderstanding, not to argue value judgments.

---

> > > > ### Author Response · Authors · 2023-11-22
> > > > **Response to Official Comment by Reviewer dCjW**
> > > >
> > > > Dear Reviewer dCjW,
> > > >
> > > > Thank you for taking the time to review our paper and provide further feedback. However, it seems that **your judgement of our paper is still based on misunderstanding** of our motivation and contribution. We would like to offer some clarifications to ensure our contributions and motivations are fully understood.
> > > >
> > > > First of all, we would like to emphasize our main contribution again, which is to investigate if the "less is more" phenomenon observed on single-modal LLMs still holds for multi-modal LLMs. In particular, we want to show if there exists a subset of the instruction dataset (as smaller as better) such that the model trained on this subset can achieve comparable performance on the evaluation tasks. Thus, **we design a learning-based method** by formulating a set of indicators for assessment and training a neural network as a data selector to fit these indicators to the genuine quality labels, which aims to identify an optimal subset for model training.
> > > >
> > > > Secondly, we would like to clarify again that **data pruning methods is hard to directly applied to this scenario**, since previous data pruning methods either rely on the model "loss" from the "training dataset" where there exists a gap to the "performance" on the "test dataset" in our problem, or require accesses to the evaluation metric which involve extensive computations and are impractical for large models. Thus, **it is unfair for data pruning methods to compare with our method** in this scenario (as they indeed empirically underperform our method).
> > > >
> > > > Moreover, we would like to clarify again that **data pruning methods have several other important differences** from our proposed method. While data pruning is a model performance driven method, we adopt a learning-based method by training a data selector, which can be applied to new datasets independent of certain models. Besides, most of data pruning methods focus on optimizing models based on the loss, while we extract the multimodal data indicators to train the selector based on MLLM performance, which manages to bridge the gap in multimodal data distribution in evaluation sets. Furthermore, data pruning aims at cutting off bad data for training, while we concentrate on identifying a minimal yet effective subset of data, offering a novel perspective in MLLM's training.
> > > >
> > > > Lastly and most importantly, it seems that your judgement is based on the misunderstanding of our motivation and contribution, and **it is unfair to directly reject our paper without providing specific reasons** (ICLR [Reviewer guideline](https://iclr.cc/Conferences/2024/ReviewerGuide#Reviewing%20instructions) states that "Clearly state your recommendation (accept or reject) with one or two key reasons for this choice."). In order to arrange **a constructive discussion** and better clear your misunderstandings, **we kindly request that you provide specific reasons, details, questions, or suggestions for further discussion.** Your input would be immensely valuable in refining our work and addressing any outstanding issues.

---

> > > > > ### Comment · Reviewer_dCjW · 2023-11-22
> > > > >
> > > > > > we kindly request that you provide specific reasons, details, questions, or suggestions for further discussion.
> > > > >
> > > > > I have provided my reasons and suggestions. You just disagree with them. I kindly request that you carefully read my comments, do the necessary literature survey, and apply the same critical evaluation to your own paper as you do to my review.
> > > > >
> > > > > > it is unfair for data pruning methods to compare with our method in this scenario (as they indeed empirically underperform our method).
> > > > >
> > > > > The authors are arguing the potential baselines (which they haven't run) are unfarily disadvantaged, therefore the comparison is impossible. First, that's their judgment, which is unsupported by any evidence.
> > > > >
> > > > > Second, assuming that is indeed true, their method should beat the data pruning methods easily, so there should be no reason not to compare against them.
> > > > >
> > > > > Lastly, they are arguing that, since they outperform some baselines, *all* baselines must be unfairly disadvantaged. Therefore, they must easily outperform *all* baselines, and there is no need to perform further comparisons. Hmmm... I feel something is not quite right but somehow could not quite pinpoint it. Maybe the meta-reviewer can help me out here.
> > > > >
> > > > > > require accesses to the evaluation metric which involve extensive computations and are impractical for large models.
> > > > >
> > > > > I'm not sure which technique the authors are referring to. Influence functions can be done using first-order gradient only. Since mini-gpt4 trains only one layer, I do not see why that's impossible to do.
> > > > >
> > > > > > data pruning aims at cutting off bad data for training, while we concentrate on identifying a minimal yet effective subset of data, offering a novel perspective in MLLM's training.
> > > > >
> > > > > That's splitting hairs. Again, assuming that argument is true, they have not compared with any coreset selection baseline such as [1]. There is an abudance of literature on that topic.
> > > > >
> > > > > [1] Xiaobo Xia, et al. Moderate Coreset: A Universal Method of Data Selection for Real-world Data-efficient Deep Learning. ICLR 2023.
> > > > >
> > > > > > The main contribution of our work is to explore the "less is more" concept in the context of multimodal Large Language Models (MLLMs).
> > > > >
> > > > > Given the abudant literature on data pruning and coreset selection, why wouldn't it be true? What is the value in showing an obvious deduction? In fact, this paper has not shown in this concept "in the context of multimodal Large Language Models (MLLMs)", but merely two MLLMs out of dozens. My judgment is that the authors have not provided a fully justified motivation. I'm sure they will disagree with me on that, and I'm not interested in convincing them otherwise.
> > > > >
> > > > > > Your input would be immensely valuable in refining our work and addressing any outstanding issues.
> > > > >
> > > > > If you value my comments as you say you do, you would run the experiments as I suggest.

---

> > ### Author Response · Authors · 2023-11-22
> > **Further Response to Reviewer dCjW (2/3)**
> >
> > Q3: Second, the authors needed manual intervention on the results of the clustering algorithm.
> >
> > A3: Thanks for your comment. **The need for manual intervention of the clustering algorithm shouldn't be considered as an issue.**
> >
> > First of all, the intervention is simple and straightforward without any complex process. This is in line with the principle of clustering algorithms and maintains the similarity of data points in each cluster.
> >
> > Secondly, the intervention is necessary. To acquire genuine quality labels which necessitate training a multimodal model on each cluster group for evaluation, we need to ensure the size of each cluter equal without letting the quantity of data impact the capabilities of the trained multimodal models.
> >
> > Moreover, the intervention is not what we focus on for this paper. There are also many same-size clustering algorithms, such as [1] and [2]. The intervention can also be replaced by them, possibly resulting in higher performance or not. However, these methods are much more complex. Our experimental results are already relatively promising. Besides, the distance of each datum from the cluster has little impact when generating quality labels, so the requirements for the cluster are not high.
> >
> > [1] Zhu, Wang, Li. Data clustering with size constraints. Knowledge-Based Systems, 2010, 23(8): 883-889.
> >
> > [2] Ganganath, Cheng, Chi. Data clustering with cluster size constraints using a modified k-means algorithm. 2014 International Conference on Cyber-Enabled Distributed Computing and Knowledge Discovery.
> >
> > Q4.1: Third, there is little understanding of what the data selection process is doing. There is no theoretical analysis nor any form of intuition. The data selector heavily relies on the length factor. It is possible that the training data are mismatched in length with the quality label datasets, so length becomes an important factor. In that case, it is hard to imagine that this would generalize to other datasets and other models.
> >
> > Q4.2: It is important for the authors to make unambiguous claim regarding their contribution. If they are claiming high performance, then perhaps it is ok to have little understanding of how their technique works, as long as performance is very high (SOTA performance, for example). If they are claiming a sound algorithm, then we do need to understand why the algorithm is sound. I am not convinced that the authors have achieved either objective.
> >
> > A4 for Q4.1 and Q4.2: Thanks for your comment. **The main contribution of our work is to explore the "less is more" concept in the context of multimodal Large Language Models (MLLMs).** Specifically, we aim to investigate whether a smaller subset of instructional data could yield comparable or superior performance in evaluation tasks, despite the differing data distributions and task types from the training tasks. This unique focus inherently differs from traditional data pruning methods, which typically optimize models based on training loss of specific training sets and models. Such methods, while valuable, may not directly translate to improved performance on diverse evaluation tasks due to potential gaps in data distributions and task types. To solve this, we employ a learning-based method using a neural network as a data selector. This selector is trained to fit indicators to genuine quality labels for optimal data selection, independent of any specific model. This distinction is crucial as it reflects our method’s unique capability to bridge the gap in data distribution in evaluation sets.
> >
> > Moreover, **we would like to clarify again that a factor with high performance does not indicates it is a shortcut**. For example, consider a classification task of determining whether there is a cow in the input image. Suppose the training data of cows always appear with the grass. If the models use a feature from either the grass (shortcut) or the cow (true feature) as its unique feature, the performance of such a model will be high. Thus, one cannot directly judge whether the grass or cow is a shortcut based on the high performance. Instead, we should test the model on other dataset, such as cows with rivers. If the model still behaves well, then the feature is not a shortcut. Similarly in our tasks,  when we use length as the unique feature, we can not conclude if length is a shortcut based on its high performance. In contrast, we should determine if length is a shortcut by testing it on various datasets with large distribution gaps such as MME. In fact, we indeed observe that selector has good performance in other tasks like MME, suggesting that length is not a shortcut but an important feature.

---

> > ### Author Response · Authors · 2023-11-22
> > **Further Response to Reviewer dCjW (3/3)**
> >
> > Q5: The same issue happens with the input to F, which has variable dimensionality. The authors have not explained why the network can accept input with different dimensions. It may work as if a convolutional network is used (or self-attention, which is basically temporal convolution) but the text says it could also be an MLP or linear regression in the ablation. Rather than explaining the reason, they verbally guarantee it. This seems to present the reviewers with a binary choice: either we take their word for it or we do not.
> >
> > A5: Sorry for the confusion. **The concatenation in the text "we concatenate these embeddings into a single composite embedding" is not along the feature dimension, but along the data dimension.** We would like to clarify again that the network actually accept input with the same dimension D, which is not K*D. When we form a cluster of K data points, these D-dimensional vectors are concatenated to create a feature vector with a shape of (K, D). In training the data selector F, we utilize these (K, D) shape vectors, which can be regarded as K data points each with a D-dimension size, just like a batch. When applied to a single data point, F operates on (1, D) feature vector for a single data point. Thus, the input shape for the data selector remains D, no matter self-attention, MLP or linear model. We have added clearer description in the revised version.

---

> ### Author Response · Authors · 2023-11-22
> **Further Response to Official Comment by Reviewer dCjW**
>
> Dear Reviewer dCjW,
>
> Thank you for your prompt and detailed feedback. **Providing specific concerns and questions is indeed an effective way to initiate an in-depth discussion**, which we truly appreciate. Regarding any concerns and misunderstandings you may have, we would like to further clarify our points as follows.
>
> First of all, on the matter of a limited number of baselines, we would like to emphasize that **there are only two existing baselines (Alpagasus[1] and Instruction Mining[2]) highly relevant to our work, and we have already included these two baselines in our initial submission**. Since the problem we are exploring on multimodal LLM is quite innovative, it results in a scarcity of related baselines. This also underscores the novelty of our work. In other words, we have included all the relevant baselines in our study.
>
> Secondly, we understand your concern about data pruning methods. Regarding the baselines for data pruning methods, we have provided an **intuitive analysis explaining why data pruning methods are hard to surpass our approach**. Contrary to your comment that we lack comparisons, we have actually selected two recent baselines as per your request, **including the most recent one** (prototypicality in 2022) from the five methods you provided. These two baselines have indeed demonstrated that data pruning methods are inferior to our data selector. **In response to your persistent inquiry about additional baselines** beyond the two data pruning methods we have already included, we are actively conducting experiments with the second most recent method (Diverse Ensembles in 2022) from the five you mentioned. We commit to promptly **updating you with the results as soon as they become available**.
>
> Thirdly, we currently **include four baselines** in our experimental results: two directly related to our research ([1], [2]) and two recent data pruning methods. We believe that this provides a comprehensive comparison framework, especially considering that **similar studies ([1], [2], [3], [4]) did not incorporate any data pruning methods in their baseline analysis**. Thus, we respectfully disagree with the comment that "the list of baselines is still rather short and only scratches the surface of the existing literature".
>
> In addition, we are referring to Data Shapley when mentioning "require accesses to the evaluation metric which involve extensive computations and are impractical for large models". Data Shapley requires plenty of iterations for model evaluation before meeting the convergence criteria. Therefore, it is not applicable for data selection in MLLMs.
>
> Lastly, and more importantly, **we respectfully disagree that our contribution of exploring the "less is more" concept in the context of MLLM is not obvious**. For your reference, we would like to draw attention to a similar but single-modal work -- **LIMA** [3]. LIMA involves simply and manually curating 1000 instructions for fine-tuning, and **they concluded that the "less is more" principle holds true for large language models**. Notably, **LIMA has gained significant impact**, accumulating 129 citations in just 6 months from its release date and receiving strong promotion from distinguished researchers such as Yann LeCun. Given the significant recognition and impact of LIMA's work, our exploration of this phenomenon in large multi-modal models adds valuable insights to the field.
>
> We hope this additional response alleviates your concerns and misunderstandings, and we are **open to further discussion** to ensure the quality and clarity of our work.
>
> [1] Chen, Li, et al. Alpagasus: Training a better alpaca with fewer data. arXiv preprint arXiv:2307.08701, 2023.
>
> [2] Cao, Kang, Sun. Instruction mining: High-quality instruction data selection for large language models. arXiv preprint arXiv:2307.06290, 2023.
>
> [3] Zhou, Liu, et al. Lima: Less is more for alignment. arXiv preprint arXiv:2305.11206, 2023.
>
> [4] Li, Yu, et al. Self-alignment with instruction backtranslation. arXiv preprint arXiv:2308.06259, 2023.

---

### Author Response · Authors · 2023-11-18
**A Summary of Paper Updates**

We thank all reviewers for the constructive suggestions, which help make this work more complete. Following their suggestions, we have made the following major updates to the paper:

Section 2: emphasize our motivation for InstructionGPT-4.

Section 2.1: clarify the manner of the generated responses.

Section 2.2: quote Appendix A.2 for detailed description of quality labels; clarify the process of dimensionality reduction.

Section 2.3: quote Appendix A.3 for details in the training and testing stages of the data selector.

Section 3.1: add explanation to K-Means++ clustering.

Section 4.4: clarify the experimental baselines; add the analysis of different data pruning methods; add the analysis of the stability and generalizability of data selector.

Section 5: add related works suggested by reviewers.

Section 6: summarize our contribution.

Appendix A: explain the reason for choosing four VQA
datasets to generate quality labels; add more discussions of the data selector in training and testing.

Appendix B: add new experiments including analysis on different selected data sizes, different data pruning methods, multiple random seeds, different multimodal LLMs and datasets to further show the superiority of our method.

---

### Meta-Review · Area_Chair_5Cdd · 2023-12-15

**Metareview:**

**Scientific Claims and Findings**:
The paper introduces InstructionGPT-4, a multimodal language model, which is built by training on the proposed method for automatically selecting high-quality diverse subsets of large multimodal instruction datasets. The authors use a small network to learn the relationship between indicator features and dataset quality, measured through loss on a validation set. Clustering is performed for diversity before data selection. The evaluation shows that the model trained on 200 examples using their data selection method outperforms models trained on all available data (MiniGPT-4) and randomly chosen examples.

**Strengths of the Paper**:

- The proposed data selection method is reasonable, and the evaluation suggests its effectiveness. It outperforms models trained on all available data and randomly chosen examples.
- The paper addresses an important challenge in finetuning Multimodal Large Language Models (MLLMs) by providing a method for selecting high-quality subsets of instruction data, contributing to improved model performance.
- The evaluation is reasonable, covering benchmarks and open-ended GPT-4-based evaluation. The authors perform ablation studies to justify their method's components, demonstrating thoughtful analysis.

**Weaknesses of the Paper**:

- Weak Experiments. The paper explores only one model/architecture, and it remains unclear how well the method would generalize to larger or different models. The baseline method MiniGPT-4 is too weak to demonstrate the effectiveness of the proposed data selection methods to attract a broader audience, its instruction tuning data quality itself is small and not strong. Please consider LLaVA-158K data that are created by GPT-4 and even larger or higher-quality dataset. The transferability of rules derived from smaller and weak data to better ones is uncertain.

- Weak Evaluation. There are at least a dozen of new benchmarks specifically proposed for MLLMs, some of them focus on real-life tasks. It will be more impactful to verify the data selection strategy by evaluating them on all of them. The current evaluation is limited. The discrepancy in performance with MiniGPT-4 on limited evaluation benchmarks/metrics in certain tasks raises concerns about the generalizability of the proposed method.

**Justification For Why Not Higher Score:**

This paper receives very intensive discussion. The idea and approach is reasonable and worth trying. However, the experiments are weak: please consider a much stronger experimental setting: bigger data and more comprehensive datasets.

**Justification For Why Not Lower Score:**

N/A

---

### Decision · Program_Chairs · 2024-01-16

Reject